# Panoramic analysis of coronaviruses carried by representative bat species in Southern China to better understand the coronavirus sphere

Yelin Han[1,2,3,4,6], Panpan Xu[1,2,3,4,6], Yuyang Wang [1,2,3,4,6], Wenliang Zhao[1,2,3,4], Junpeng Zhang[5], Shuyi Zhang[5], Jianwei Wang[1], Qi Jin[1,3,4] ✉ & Zhiqiang Wu [1,2,3,4] ✉

Bats, recognized as considerable reservoirs for coronaviruses (CoVs), serve as natural hosts for several highly pathogenic CoVs, including SARS-CoV and SARS-CoV-2. Investigating the bat CoV community provides insights into the origin for highly pathogenic CoVs and highlights bat CoVs with potential spillover risks. This study probes the evolution, recombination, host range, geographical distribution, and cross-species transmission characteristics of bat CoVs across China and its associated CoVs in other regions. Through detailed research on 13,064 bat samples from 14 provinces of China, 1141 CoV strains are found across 10 subgenera and one unclassified Alpha-CoV, generating 399 complete genome sequences. Within bat CoVs, 11 new CoV species are identified and 425 recombination events are detected. Bats in southern China, particularly in Yunnan province, exhibit a pronounced diversity of CoVs. Limited sampling and low detection rates exist for CoVs in *Myotacovirus*, *Nyctacovirus*, *Hibecovirus*, *Nobecovirus* in China. The genus *Myotis* is highlighted as a potential ancestral host for Alpha-CoV, with the genus *Hipposideros* suggested as a likely progenitor host for bat-associated Beta-CoV, indicating the complexity of cross-species transmission dynamics. Through the comprehensive analysis, this study enriches the understanding of bat CoVs and offers a valuable resource for future research.

With the encroachment of wildlife habitat due to human activity, human-wildlife overlap and contact increase the risk of viruses jumping from wild animals to humans and potentially causing an outbreak of emerging infectious diseases (EIDs). Since the 21st century, humans have faced severe threats from three EIDs caused by coronaviruses (CoVs): the Severe Acute Respiratory Syndrome (SARS) epidemic triggered by SARS-CoV in 2002, the Middle East Respiratory Syndrome (MERS) epidemic triggered by MERS-CoV in 2012, and the Corona Virus

[1]NHC Key Laboratory of Systems Biology of Pathogens, Institute of Pathogen Biology, Chinese Academy of Medical Sciences & Peking Union Medical College, Beijing, China. [2]Key Laboratory of Respiratory Disease Pathogenomics, Chinese Academy of Medical Sciences & Peking Union Medical College, Beijing, China. [3]Key Laboratory of Pathogen Infection Prevention and Control (Peking Union Medical College), Ministry of Education, Beijing, China. [4]State Key Laboratory of Respiratory Health and Multimorbidity, Chinese Academy of Medical Sciences, Beijing, China. [5]College of Animal Science and Veterinary Medicine, Shenyang Agricultural University, Shenyang, China. [6]These authors contributed equally: Yelin Han, Panpan Xu, Yuyang Wang. ✉e-mail: zdsys@vip.sina.com; wuzq2009@ipbcams.ac.cn

Disease 2019 (COVID-19) pandemic triggered by SARS-CoV-2 at the end of 2019[1–3]. Notably, each of these viruses (SARS-CoV, MERS-CoV, and SARS-CoV-2) that led to an epidemic or pandemic, has been proven to originate from bats[3–7]. Since bats are the primary reservoir hosts for members of the genera *Alphacoronavirus* (Alpha-CoV) and *Betacoronavirus* (Beta-CoV)[8–10], a diverse range of CoVs that bats carry present a continuous threat to the present and future of human society.

In accordance with the International Committee on Taxonomy of Viruses (ICTV) (https://ictv.global.), CoVs can be divided into four genera: Alpha-CoV, Beta-CoV, *Deltacoronavirus*, and *Gammarcoronavirus*. The first two genera mainly originate from bats[8,9]; Alpha-CoV includes 15 subgenera, 10 of which are naturally hosted by bats. Beta-CoV contains five subgenera, four of which are naturally hosted by bats. Currently, seven known CoVs can infect humans and cause mild to severe respiratory diseases. Among them, HCoV-229E[11] and HCoV-NL63[12] belong to the subgenera *Duvinacovirus* and *Setracovirus* of Alpha-CoV, respectively; HCoV-OC43[13], HCoV-HKU1[14], SARS-CoV, MERS-CoV, and SARS-CoV-2 belong to the *Embecovirus*, *Merbecovirus*, and *Sarbecovirus* of Beta-CoV, respectively. In addition to infecting humans and bats, CoVs from the two genera can also infect many other mammals such as pigs, cattle, horses, dogs, hedgehogs, and cats[8]. The unique viral replication mechanism of CoVs makes them recombine frequently[6,15–18], hence the genome of CoVs demonstrates significant plasticity and may facilitate potential cross-species transmission[19]. Such characteristics could potentially be linked to the outbreaks of SARS, MERS, and COVID-19.

The rich diversity of bat CoVs within China may be intricately tied to the abundant species and their widespread geographical distribution[20,21]. At present, among the 14 known subgenera of bat CoVs, 10 of these subgenera can be found in bats from China, including *Decacovirus*, *Minunacovirus*, *Myotacovirus*, *Nyctacovirus*, *Pedacovirus* and *Rhinacovirus* of Alpha-CoV, and *Hibecovirus*, *Merbecovirus*, *Nobecovirus* and *Sarbecovirus* of Beta-CoV[9]. The discovery of SARS-CoV related CoVs (SARSr-CoVs) and SARS-CoV-2 related CoVs (SC2r-CoVs) from horseshoe bats (family Rhinolophidae) in East Asia, Southeast Asia, and other areas has put bat CoVs in these regions under heightened scrutiny[22–28]. Likewise, bat species from the families Vespertilionidae and Craseonycteridae have been identified as hosts of MERS-CoV related CoVs (MERSr-CoVs). Yet, despite several specific bat CoVs identified by different research teams in China, a comprehensive survey for bat CoVs remains a challenge.

In this work, we undertook a detailed virome analysis to delve deeply into the CoV diversity in 13,064 bat samples spanning 54 bat species, 19 genera, and seven families from China between 2016 and 2021. Our findings build on the previous study[29], which reported 146 bat CoVs in *Sarbecovirus*. Including these, our research identified a total of 1141 CoV strains, and from these, we successfully obtain 399 complete genome sequences. Our investigation explored the evolutionary, recombination, host range, geographical distributions, and cross-species transmission characteristics of bat CoVs. Significantly, this study sheds light on the potential evolutionary processes of highly pathogenic CoVs, such as SARS-CoV-2, and evaluates the zoonotic potential of important bat CoVs. Further, our analysis refines the taxonomy and co-evolutionary features of bat Alpha- and Beta- CoVs, bolstering the foundational understanding of bat CoVs.

## Results

### Bat sample collection
From August 2016 to July 2021, a total of 13,064 oral and anal swabs were collected from bats in 14 provinces across China. Of these, 4755 samples were collected following the COVID-19 outbreak. Sampling sites covered the hotspots of bats carrying Alpha-CoV and Beta-CoV in China[9,29] (Supplementary Data 1 and Data 2), including the Yunnan province ($n = 2487$), Sichuan province ($n = 892$), Guangdong province ($n = 1991$), Guizhou province ($n = 356$), Hainan province ($n = 587$), Hunan province ($n = 286$), Jiangxi province ($n = 508$), Anhui province ($n = 36$), Zhejiang province ($n = 565$), Fujian province ($n = 161$), Liaoning province ($n = 202$) Guangxi Zhuang Autonomous Region ($n = 3257$) and Chongqing City ($n = 64$) (Fig. 1 and Supplementary Data 1). The sampled bats belonged to a wide range of species, including 54 identified species and 4 species that are yet to be determined, belonging to 19 genera and seven families. Bats from the genus *Rhinolophus* ($n = 4270$), genus *Myotis* ($n = 1949$), and genus *Hipposideros* ($n = 1996$) had sampling advantages, accounting for 62.88% of the total samples. In addition, most samples were collected from *Rhinolophus sinicus* (*R. sinicus*) ($n = 1415$), *R. pusillus* ($n = 1066$), *Scotophilus kuhlii* (*S. kuhlii*) ($n = 1045$), *Tylonycteris pachypus* (*T. pachypus*) ($n = 1014$), *H. larvatus* ($n = 958$) and others. A few samples originated from *R. paradoxolophus* ($n = 1$), *Myotis formosus* (*M. formosus*) ($n = 1$), *M. nipalensis* ($n = 1$), *M. rufoniger* ($n = 1$) and *Pipistrellus ceylonicus* (*P. ceylonicus*) ($n = 1$) (Supplementary Data 1). This comprehensive sampling strategy provides a robust foundation for understanding the diversity and distribution of bat CoVs in China.

### CoV screening
Based on species, sampling location, sampling time, and other related factors, bat samples were combined into 372 pools, and then libraries of nucleic acids were constructed for high-throughput sequencing and PCR screening (Supplementary Fig. 1) A total of approximately 764 GB of clean data was obtained. A total of 571,486,589 reads related to viral protein sequences in the NR library, including 28,014,438 reads related to CoV, and the number of CoV-related reads in each pool ranged from 0 to 2,292,787 reads. Combined screening showed that 113 pools were Alpha-CoV positive alone, 64 pools were Beta-CoV positive alone and 22 pools were combined Alpha-CoV and Beta-CoV positive. In order to comprehensively describe the overall characteristics of bat CoVs, we further screened all 4761 single samples involving the CoV-positive pool. Finally, a total of 1141 CoV-positive samples were identified, including 146 sarbecovirus-positive samples identified previously (Supplementary Data 2). Based on the simulation of bat distribution in China and the projection of identified positive coronavirus sampling points in this study, we observed a general trend where areas with abundant bat populations were associated with a higher likelihood of coronavirus detection. (Supplementary Fig. 2)

The 1141 CoV positives belonged to 10 subgenera of CoV and a group of unclassified Alpha-CoV. These included 7 subgenera under Alpha-CoV, namely *Pedacovirus* ($n = 300$), *Rhinacovirus* ($n = 151$), *Minunacovirus* ($n = 134$), *Decacovirus* ($n = 129$), *Myotacovirus* ($n = 73$), *Nyctacovirus* ($n = 10$) and unclassified Alpha-CoV ($n = 8$). In addition, 4 subgenera belonged to Beta-CoV, namely *Merbecovirus* ($n = 138$), *Sarbecovirus* ($n = 161$), *Nobecovirus* ($n = 36$), and *Hibecovirus* ($n = 1$). Excluding the previously reported *Sarbecovirus*[29], 371 strains as representative strains were selected for whole-genome sequencing, yielding 330 complete sequences. Out of these, 240 sequences were of Alpha-CoV including *Pedacovirus* ($n = 84$), *Decacovirus* ($n = 48$), *Rhinacovirus* ($n = 42$), *Minuacovirus* ($n = 33$), *Myotacovirus* ($n = 20$), *Nyctacovirus* ($n = 9$), and unclassified Alpha-CoV ($n = 4$), respectively. In addition, 90 sequences were of Beta-CoV, comprising *Merbecovirus* ($n = 64$), *Nobecovirus* ($n = 16$), *Sarbecovirus* ($n = 9$) and *Hibecovirus* ($n = 1$). Cumulatively, 399 whole-genome sequences of CoVs were determined, including 69 *Sarbecovirus* sequences previously studied[29] (Fig. 1 and Supplementary Data 2).

Differences were found in the detection rate of bat CoVs among different bat families and species, according to the results of CoV screening. It was noted that CoVs were detected in 36 bat species across 14 provinces in China (Fig. 2 and Supplementary Data 3). Significant differences were found in the detection rates of certain subgenera of CoVs across different bat families (Supplementary Data 4). *Decacovirus* was detected in bats of the Hipposideridae family at a rate

significantly higher than in Vespertilionidae bats, while *Nobecovirus* was found in bats of the Pteropodidae family at a rate significantly higher than that in bats from other families. Further, the detection rate of *Rhinacovirus* in bats of the Rhinolophidae family was considerably higher than that in bats of the Hipposideridae and Vespertilionidae families. Similarly, the detection rate of *Sarbecovirus* in bats of the Rhinolophidae family was notably higher than that in bats of the Vespertilionidae family.

Among the detected CoVs, the most diverse at the CoV subgenus level were found in the family Vespertilionidae, which included *Decacovirus*, *Minunacovirus*, *Myotacovirus*, *Nyctacovirus*, *Pedacovirus*, *Merbecovirus*, and an unclassified Alpha-CoV. CoVs of the *Hibecovirus* were exclusively detected in *H. pratti* from Hubei province, while CoVs of the *Nobecovirus* were solely detected in *Eonyctris spelaea* and *Rousettus leschenaultia* from the Yunnan province. Detection rates of viruses differed among different bat species (Supplementary Data 5). In

general, the detection rate of coronaviruses in bats of the Rhinolophidae family was higher than in some other species. Among various species of the Rhinolophidae family, *R. affinis*, *R. pusillus*, and *R. sinicus* exhibited relatively high detection rates. Moreover, the detection rate of *P. abramus* was notably higher than some bat species in the genus *Myotis*.

Due to the difference in the sampling volume from the different bat species, this study could not obtain a reliable positive rate of CoV infection for bat species with a lower sampling volume, but it is a reference value for the positive rate of CoV infection for bat species with wide distribution and high species richness in China, such as the *R. sinicus*, which had the largest sampling volume. The positive rates of *Sarbecovirus* and *Rhinacovirus* ranged from 0.01% to 11.10%, from 0.01% to 30.00% respectively, and the positive rates in different provinces varied between 0.01% and 64.70% (Supplementary Data 3).

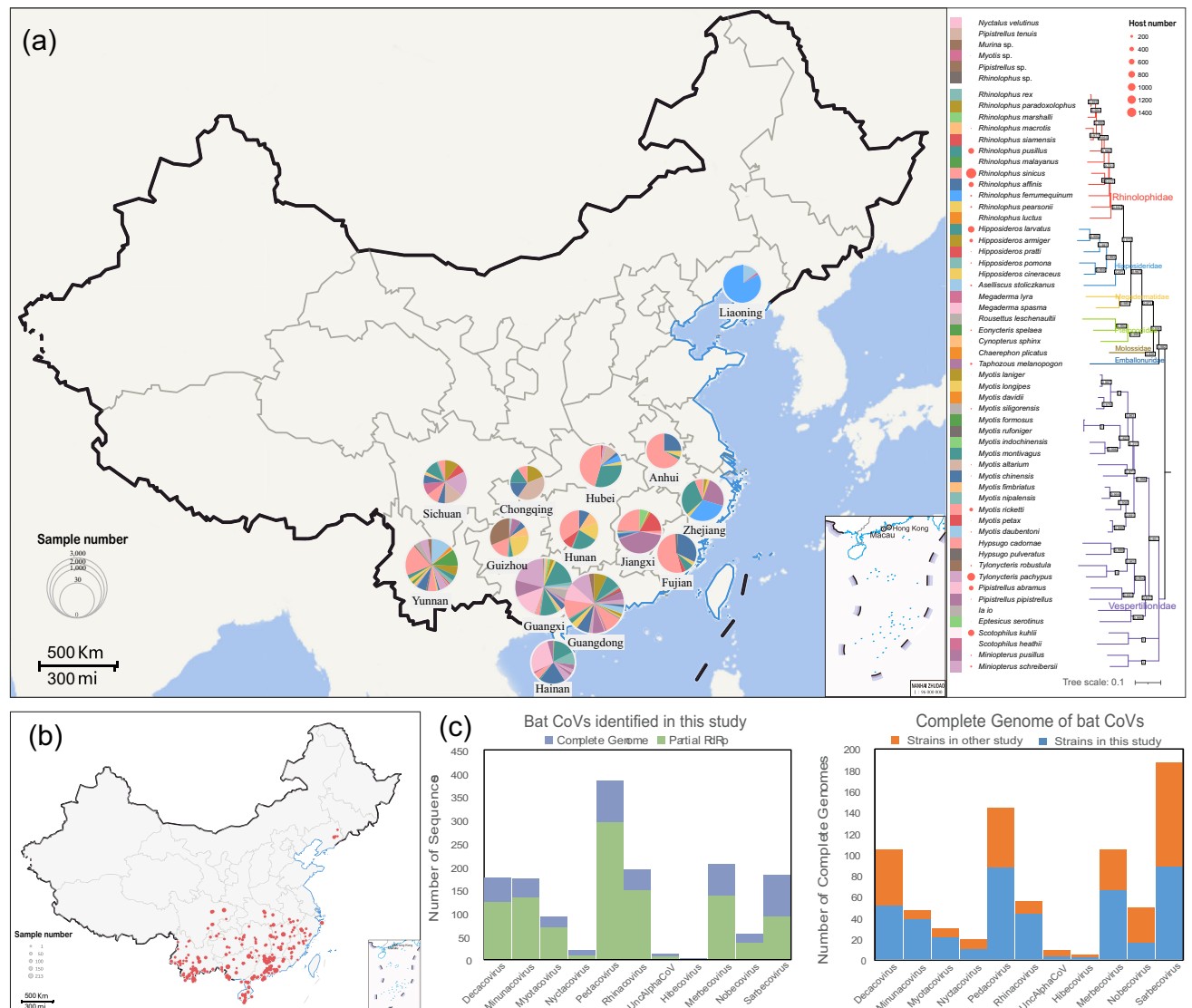

**Fig. 1 | Overview of bat sampling and bat coronaviruses identified in this study.** **a** Bat sampling in different provinces. The pie chart on the left side of the map illustrates the proportion of bat species sampled in 14 different provinces in China. The colors in the pie chart represent different species, consistent with the colors assigned to branches on the evolutionary tree based on cytb. The branches of the evolutionary tree are color-coded according to the bat's family (Supplementary Data 2). Map data were retrieved from Tianditu (https://www.tianditu.gov.cn/) (**b**) Sampling locations and numbers in this study. The sampling points in this study are

marked on the map as red dots, with the size of the dots proportional to the number of samples collected (Supplementary Data 2). Map data were retrieved from zenodo (DOI: 10.5281/zenodo.4167299) (**c**) Overview of identified Strains in this study. The stacked bar chart represents the number of strains identified in this study based on RDRP and complete genomes. It also compares the number of complete genomes identified in this study with those identified by others (Supplementary Data 2 and Data 6).

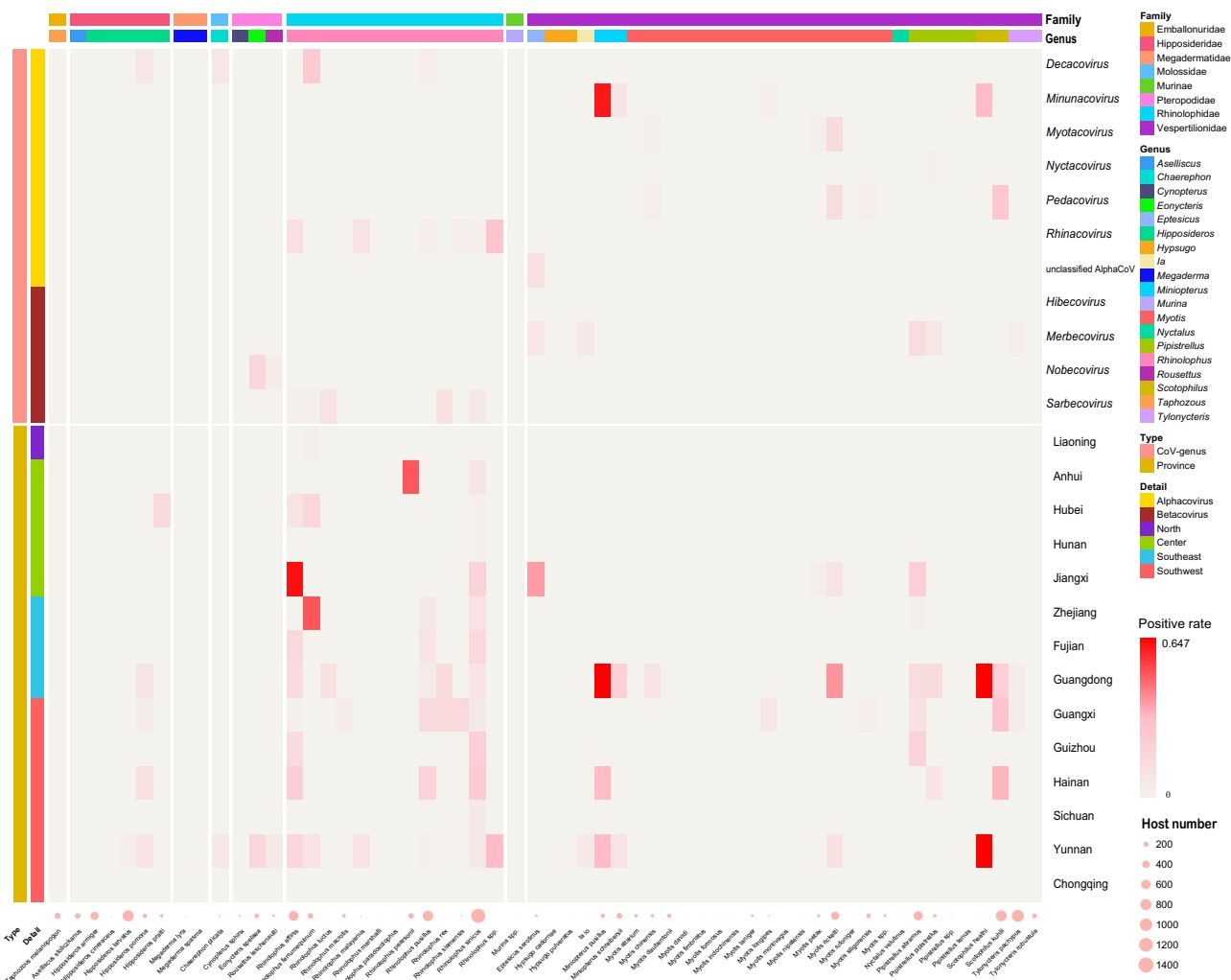

**Fig. 2 | Positive rate of bat coronavirus in different bat species and provinces in this study.** Positive rates ranged from low to high corresponding colors from light to dark red. The pink circles at the bottom represent the number of hosts in this study and the size of the circles is proportional to the number of samples (Supplementary Data 3).

## Classification of novel CoVs and evaluation the methods of screening CoVs based on partial RdRp

To gain a deeper understanding of the evolution, geographical distribution, host range, and recombination of bat CoVs in China as well as their related CoVs in other countries or regions, additional data were incorporated from GenBank (https://www.ncbi.nlm.nih.gov/genbank/), Global Initiative on Sharing All Influenza Data (GISAID: https://www.gisaid.org/) and National Genomics Data Center (NGDC: https://ngdc.cncb.ac.cn/). This included 5181 bat CoV sequences comprising 4698 RdRp partial sequences, 405 whole-genome sequences, and 78 partial genome sequences. Furthermore, to establish a more comprehensive taxonomy for Alpha-CoVs and Beta-CoVs and to depict more intricately the relationships between bat CoVs and those found in other animal species, the dataset was supplemented with 91 representative sequences. This subset included reference sequences (refseq) of non-bat CoVs in Alpha- and Beta- CoV genera, along with CoVs found in various animal species within the same subgenera as bat CoVs, including 88 whole-genome sequences and 3 partial genome sequences (GenBank, GISAID, and NGDC accession numbers and detailed information are provided in Supplementary Data 6). Building upon the CoV sequences identified in this study and those amassed from public databases, we organized these sequences into three distinct yet intersecting datasets: Dataset 1, Dataset 2, and

Dataset 3 (The sequence lists for each of these datasets can be found in Supplementary Data 2 and Data 6). The specific application of these datasets in our subsequent analyses was tailored to meet the varying requirements of each individual analytical component of our research.

Utilizing Dataset 1, analysis of amino acid sequences of the seven conserved replicase domains (CRDs) in ORF1ab of bat CoVs revealed that, except for the subgenus and species of CoVs that had been classified, bat CoVs had formed two novel evolutionary lineages at the level of the CoV subgenus, which were designated as Bat Alphacoronavirus new lineage 1 (BatAlpha_NL1) and Bat Betacoronavirus new lineage 11 (BatBeta_NL11). In addition, nine novel evolutionary lineages at the CoV species level were discerned, designated as Bat *Decacovirus* new lineage 2 (BatDeca_NL2), Bat *Decacovirus* new lineage 3 (BatDeca_NL3), Bat *Decacovirus* new lineage 4 (BatDeca_NL4), Bat *Decacovirus* new lineage 5 (BatDeca_NL5), Bat *Nyctacovirus* new lineage 6 (BatNycta_NL6), Bat *Nyctacovirus* new lineage 7 (BatNycta_NL7), Bat *Pedacovirus* new lineage 8 (BatPeda_NL8), Bat *Pedacovirus* new lineage 9 (BatPeda_NL9), and Bat *Nobecovirus* new lineage 10 (BatNobe_NL10) (Fig. 3a). The amino acid sequence identity between these novel lineages and the classified CoV species within the seven CRDs was <90% (Supplementary Data 7). Of these, the bat CoVs in China are associated with six novel evolutionary lineages, with the strains detected in this study appearing in four of these lineages, specifically,

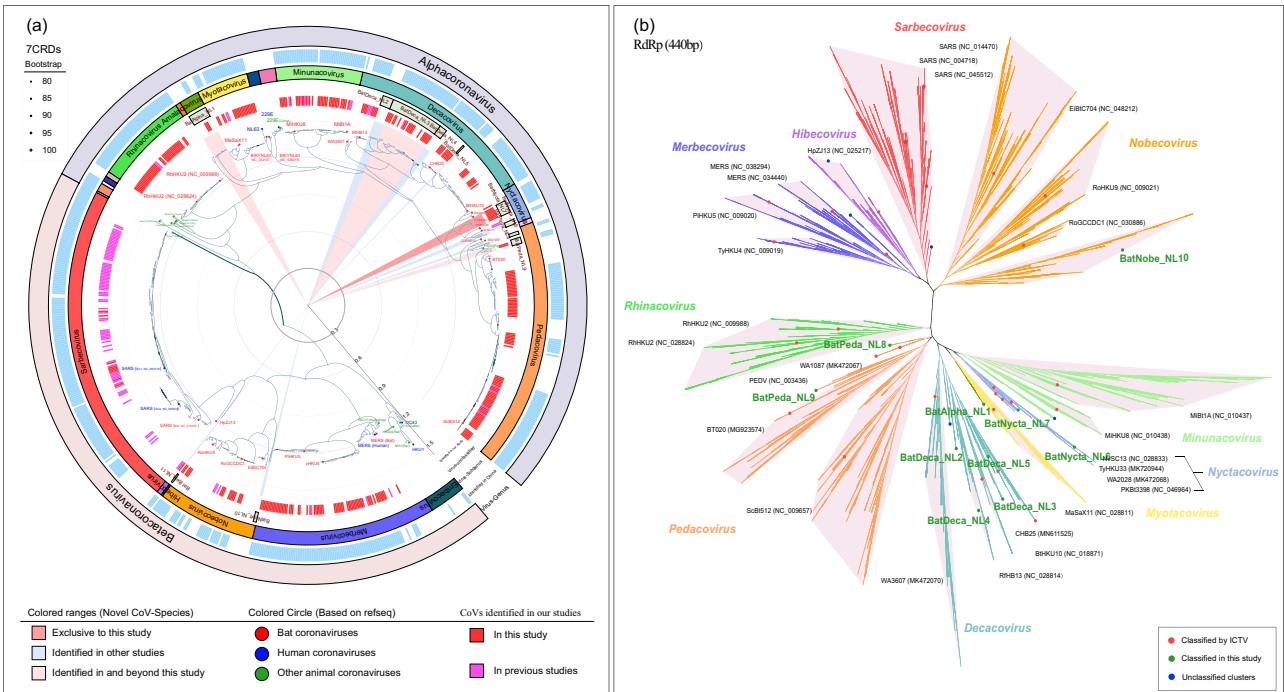

**Fig. 3 | Taxonomic phylogenetic tree based on seven conserved replicase domains in ORF1ab of bat CoVs and Phylogenetic analysis of 10 subgenera and an unclassified-Alpha-CoVs based on RdRp(440 bp). a** The trees were constructed by the maximum likelihood method using appropriate models (LG + F + I + G4) based on iqtree. The external color strips range from inner to outer represent the CoVs identified in our studies, virus-unclassified, virus-subgenus, CoVs identified in China, and virus-genus. The classified CoV species have been marked in the corresponding evolutionary branch. **b** Nucleotide sequence phylogenetic trees of the RdRp gene in different subgenera use different colors as background colors. The reference strains of each subgenus are dotted and annotated in words.

BatAlpha_NL1, BatDeca_NL3, BatNycta_NL7, and BatPeda_NL9. Among them, CoVs in BatNycta_NL7 were uniquely identified in this study. By analyzing datasets from public data and our newly identified CoVs, 14 of the 20 subgenera of Alpha- and Beta- CoVs have been associated with bats, with a few remaining unclassified. Despite the identification of a large number of bat CoVs in this study, it did not result in an expansion of the subgenus range of bat CoVs in China. Predominantly, bats in China were found to harbor 10 subgenera of CoVs and a small number of unclassified Alpha-CoVs.

The evolutionary tree, constructed using 6,658 RdRp (-440 bp) sequences (Dataset 2), demonstrated that among the 11 subgenera (including one unclassified Alpha-CoV BatAlpha_NL1) of CoV associated with bats in China, beyond the CoV species classified by ICTV and the novel evolutionary lineage discerned in this study, there remained several unclassified CoVs in the subgenera *Decacovirus*, *Minunacovirus*, *Merbecovirus*, *Hibecovirus*, and *Sarbecovirus*. As only small fragments of RdRp (-440 bp) were identified, they could not be classified based on the CoV classification criteria (Fig. 3b and Supplementary Data 6).

In the comprehensive analysis of CoVs, Dataset 1 comprising 475 sequences for Alpha-CoVs and 408 for Beta-CoVs—mirroring those utilized in the 7CRDs tree—was deployed. Phylogenetic trees, specifically for Alpha- and Beta- CoVs, were constructed using partial RdRp (-440 bp) and other genomic regions, such as ORF1ab, ORF1a, OEF1b, S, E, M, and N (Supplementary Fig. 3a and b, Supplementary Fig. 4 and Supplementary Fig. 5). Upon comparing the clustering trends within the phylogenetic trees, constructed from partial RdRp and 7CRDs, in conjunction with those from ORF1ab, ORF1a, and OEF1b, a significant degree of congruence was observed at the levels of CoV Subgenus and species. Specifically, perfect congruence was noted at the CoV Subgenus level, while basic congruence was observed at the CoV species level. Variations were only detected within BatPeda_NL8 and BatPeda_NL9, which could potentially be attributed to the inclusion of new CoV species within these two novel lineages. However, when comparing with trees derived from S, E, M, and N regions, a divergence was observed, suggesting potential limitations in using partial RdRp for examining the genomic structural diversity of CoVs. These findings underscore the reliability of partial RdRp sequencing for the identification of CoVs. Yet, the observed inconsistencies reiterate the importance of obtaining whole-genome sequences in the study of CoV diversity.

## Genomic structure and recombination events

The genomic structure of CoVs identified in this study, as well as those from 11 subgenera within the Alpha-CoV and Beta-CoV genera, was evaluated. Full genome lengths ranged from 26,956 bp to 31,491 bp, with *Hibecovirus* and *Rhinacovirus* exhibiting the longest and shortest genomes, respectively (Supplementary Fig. 6). Despite a uniform genomic organization across the 11 subgenera (5′UTR - ORF1ab polyprotein (ORF1ab) - spike protein (S) - envelope protein (E) - membrane glycoprotein (M) – nucleocapsid phosphoprotein (N) – 3′UTR-poly (A) tail), significant variations were observed in the distribution of accessory proteins (Fig. 4). In Alpha-CoV, accessory proteins were mainly located between the S and E proteins (ORF3), and between the N protein and the 3′UTR-poly (A) tail (ORF7). However, ORF7 was only detected in *Decacovirus*, *Myotacvirus*, *Rhinacovirus* and unclassified Alpha-CoV. In the case of Beta-CoV, accessory proteins were universally found between the S and E proteins, though their distribution displayed unique characteristics in each subgenus. *Nobecovirus* also contained accessory proteins primarily between the N protein and 3′ UTR-poly (A) tail (ORF7), with the *Rousettus bat coronavirus GCCDC1* (RoGCCDC1) strain exhibiting a distinctive insertion sequence named p10. Within the *Sarbecovirus* subgenus, accessory proteins were also found between the M and N proteins, with three accessory proteins, ORF6, ORF7 and ORF8, being identified. Accessory proteins in the *Merbecovirus* subgenus were exclusively situated between the S and E proteins, including ORF3, ORF4, and ORF5. The *Hibecovirus* subgenus uniquely harbored accessory proteins between ORF1ab and the S

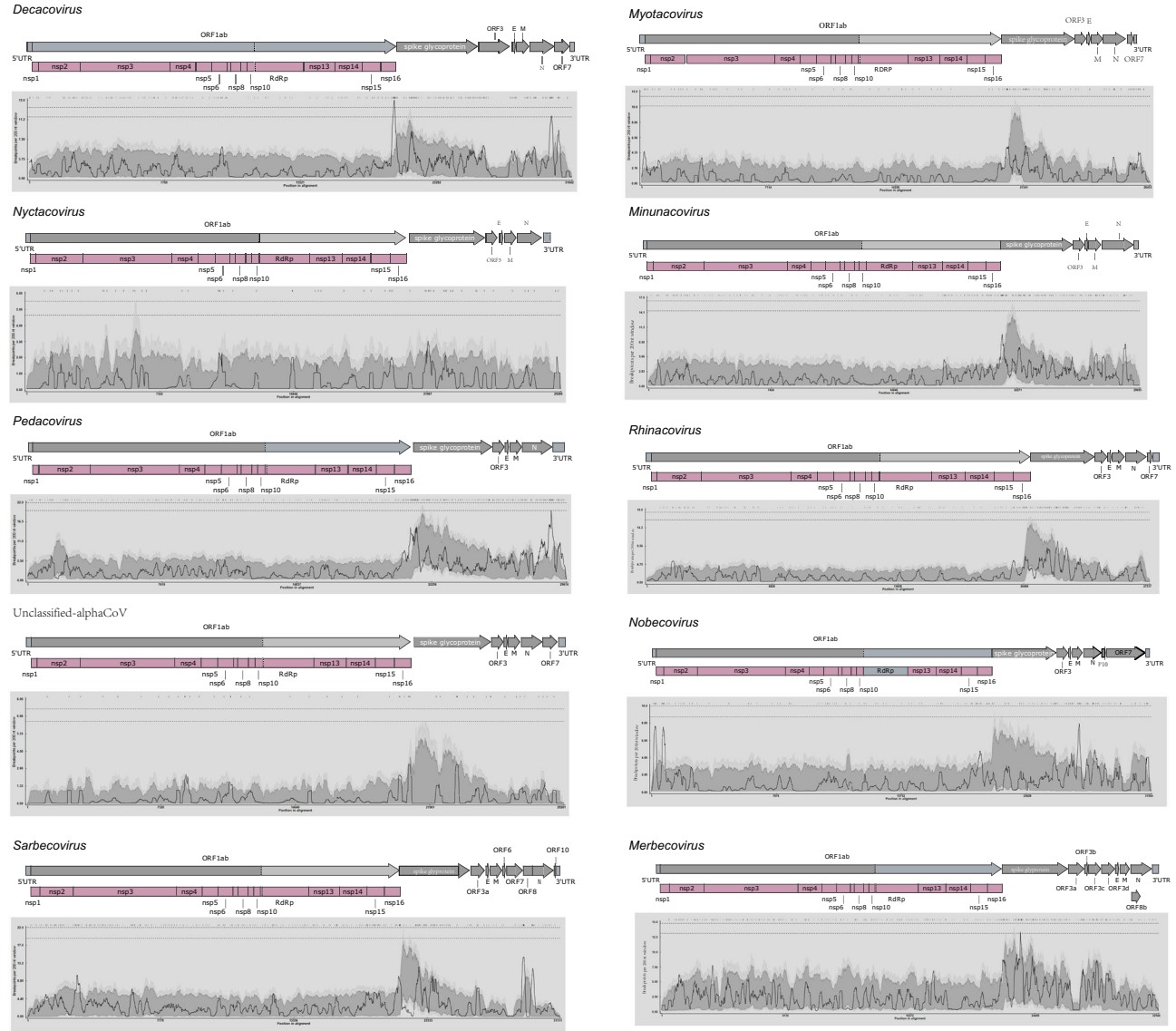

**Fig. 4 | The genome structure and recombination breakpoint plot.** Above the breakpoint plot is the corresponding genome structure plot. The black vertical lines annotated above represent the positions of detectable recombination breakpoints. The dashed lines above and below represent the 99% and 95% confidence thresholds, respectively, for the global hot-spot test. The gray and black backgrounds below represent the 99% and 95% confidence intervals, respectively, for the local hot/cold-spot test. The solid black line represents the count of breakpoints within a moving window of 200 nucleotides. The shaded area above the shadow region indicates recombination hotspots, while the shaded area below indicates recombination cold spots.

protein, as well as between the M and N proteins, commonly known as ORF7 and ORF8.

A recombination analysis was undertaken on 10 subgenera and one unclassified Alpha-CoV related to bat CoVs in China (Fig. 4), utilizing Dataset 3 for this part of the study. As a result, a total of 425 recombination events were identified across various subgenera, with the exception of *Hibecovirus*, where no recombination events were detected (Supplementary Data 8). In light of the numerous recombination events within the nine subgenera of coronaviruses and one unclassified Alpha-CoV BatAlpha_NL1, a manual curation of the breakpoints inferred by RDP5 was performed. The Breakpoint Distribution Plot[30] and permutation-based testing in RDP5 were utilized to examine the significant clustering of recombination events of the genome. This was indicative of the presence of recombination hot spots or cold spots. It was observed that *Nyctacovirus* and Unclassified Alpha-CoV did not display distinct recombination hotspots or cold spots. On the other hand, *Decacovirus*, *Pedacovirus*, *Myotacovirus*, *Minunacovirus*, *Rhinacovirus*, *Merbecovirus*, and *Sarbecovirus*

exhibited recombination hotspots at the junction of the S protein and ORF1ab. In addition, *Minunacovirus* also showed recombination hotspots in the E protein, M protein, and N protein regions. *Rhinacovirus* exhibited recombination hotspots in the S2 region, while *Nobecovirus* manifested recombination hotspots at the end of ORF1b and in the M protein region. Beyond the S region, *Myotacovirus* displayed recombination hotspots in the N protein region and ORF7. Likewise, *Pedacovirus* demonstrated recombination hotspots in the N protein region, excluding the S protein region.

## Evolutionary, host range, distribution, and recombination characteristics of *Alphacoronavirus* in bats

The *Decacovirus* currently contains four recognized species, including *Bat coronavirus HKU10* (BtHKU10), *Rhinolophus ferrumequinum alphacoronavirus HuB-2013* (RfHB13), *Alphacoronavirus WA3607* (WA3607) and *Alphacoronavirus CHB25* (CHB25). Furthermore, taxonomic analysis based on seven CRDs has revealed four evolutionary lineages of undetermined species within *Decacovirus*, denoted as

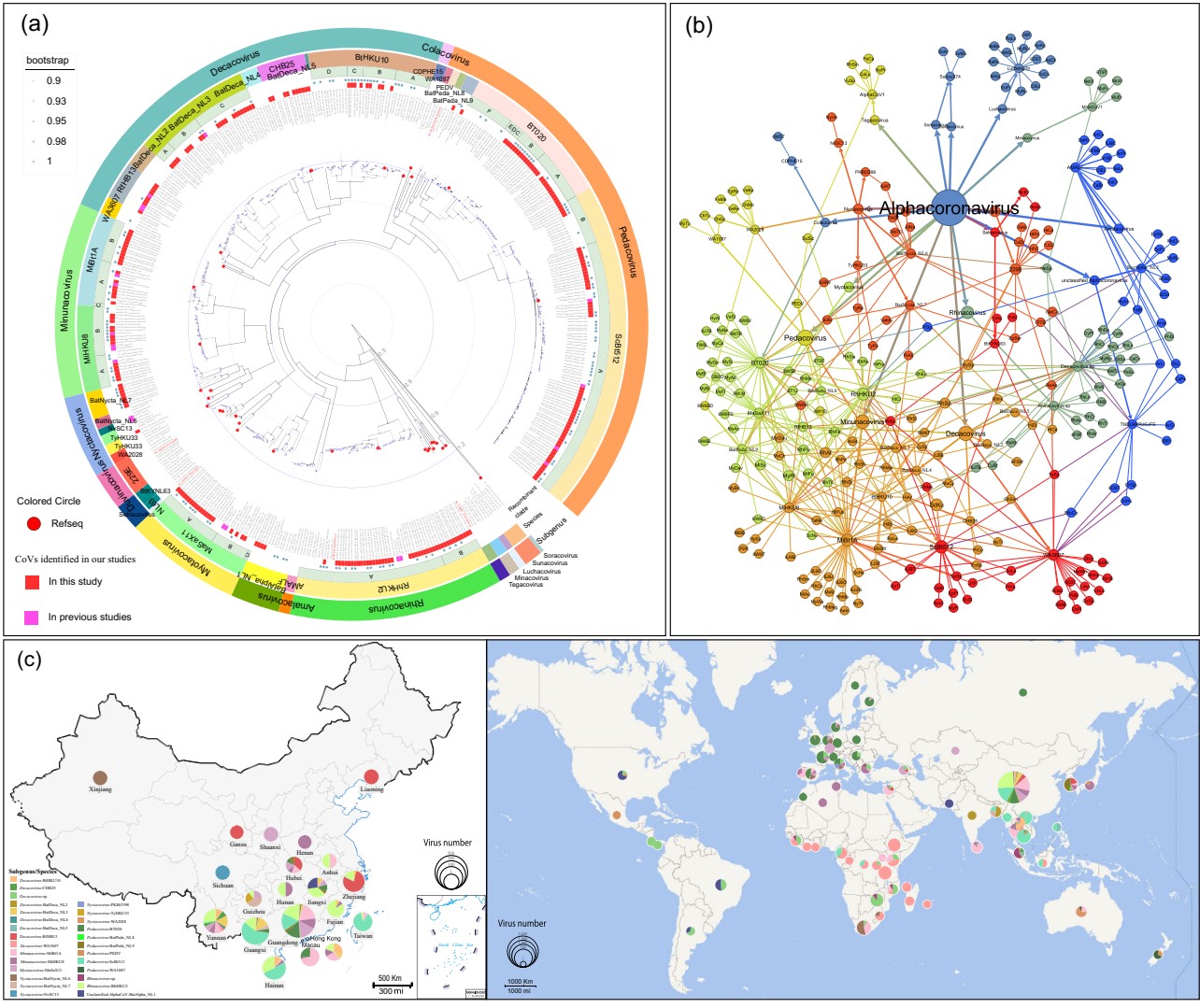

**Fig. 5 | Evolution, host range and distribution of coronaviruses in *alphacoronavirus*. a** The phylogenetic tree is based on the ORF1ab. The tree was constructed by the maximum likelihood method using appropriate models (GTR + I + G) based on iqtree. The viral species of each subgenus and their corresponding labels are present on the right of the phylogenetic tree. In addition, we have annotated the identified recombinant sequences, which were confirmed through RDP detection and manual corrections. These recombinant sequences have been marked with blue asterisks. **b** Network based on virus in *alphacoronavirus* and the corresponding host. In the network, nodes represent the genus, subgenus, species, and hosts of viruses, with node size proportional to the number of viruses. We have employed abbreviations to show the viral species and hosts, with their corresponding full names available in Supplementary Data 2 and Data 6. Lines are drawn between virus genus and its subgenus, subgenus and its species, and between each virus species and its corresponding hosts. To distinguish between the three levels of connections, the line widths decrease in the order from genus to subgenus to species. Running Community Detection named Modularity in the statistics page yields a Modularity Class value for each node, with node color ranking based on the Modularity class. The line colors change with the node colors. **c** The distribution and proportion in China and the world of each virus species in subgenus of *alphacoronavirus*. The different colors in the pie chart represent different viral species. The size of each slice in the pie chart is proportional to the total number of viruses identified in that region. World map data were retrieved from Tianditu (https://www.tianditu.gov.cn/) and China map data were from https://doi.org/10.5281/zenodo.4167299.

BatDeca_NL2 to BatDeca_NL5 (Fig. 3a). CoVs related to WA3607 (WA3607r-CoVs) were primarily identified in the Molosidae family, with a geographical distribution spanning Asia, Oceania, and Africa. The CoVs related to RfHB13, BatDeca_NL3, and BatDeca_NL4 were predominantly found in the *Rhinolophus* genus, displaying a broad distribution in Asia, particularly in China. CoVs of BatDeca_NL2 were largely observed in *Megaderma lyra* from Bangladesh. CoVs related to CHB25, BatDeca_NL5, and BtHKU10 were primarily detected in the genus *Hipposideros*, which are distributed in China and other Southeast Asian countries (Fig. 5).

Phylogenetic analysis founded on ORF1ab demonstrated that the 48 newly identified CoVs possessing complete genomes within *Decacovirus* were affiliated with BtHKU10 (*n* = 9), RfHB13 (*n* = 8), BatDeca_NL3 (*n* = 21), and CHB25 (*n* = 10) (Fig. 5a). CoVs related to BtHKU10

(BtHKU10r-CoVs) were primarily discovered in *H. pomona*, spanning several southern provinces of China, including Guangdong, Guangxi, Hainan, Yunnan, and others. Remarkably, BtHKU10r-CoVs diverged into four distinct clades (A to D) exhibiting significant differences. CoVs associated with RfHB13 were mainly found in *R. ferrumequinum*, widely distributed across central and northern China. BatDeca_NL3 related CoVs were primarily distributed in Yunnan, Guangxi, Hainan, Guangdong, and other southern provinces of China, yet segregated into three distinct clades (A to C) based on the host species. Predominant hosts of BatDeca_NL3-A and BatDeca_NL3-B were *R. affinis* and *R. sinicus*, while the host range for BatDeca_NL3-C was broader, including various *Rhinolophus* species, *M. muricola*, and *H. cineraceusi*. CoVs associated with CHB25 were mainly found in *H. larvatus* and *H. armiger*, primarily spanning Yunnan, Guangxi, and Guangdong

provinces. CoVs from WA3607, BatDeca_NL2, BatDeca_NL4, and Bat-Deca_NL5 were chiefly found outside China, not covered in this study (Fig. 5b and c, Supplementary Data 9).

Investigating the recombination events within the *Decacovirus* yielded significant findings. Specifically, 28 distinct recombination sequences were elucidated, implicating RfHB13, RhBatL3, BtHKU10, and WA3607 (Fig. 5a). Recombination events primarily occur within species, with a big proportion of recombination events observed in BtHKU10 intraspecies recombination, followed by BatDeca_NL3 intraspecies recombination (Supplementary Data 8). While no recombination sequences were identified among BatDeca_NL2-Bat-Deca_NL5, an insightful correlation was drawn from the constructed phylogenetic trees based on ORF1a, ORF1b, and structural proteins (Supplementary Fig. 4). Intriguingly, the BatDeca_NL2r-CoVs demonstrated a closer phylogenetic affinity with the *Rhinacovirus* in the S protein region, deviating from other members of the *Decacovirus*. A similar trend was observed for WA3607, which aligned more closely with the *Minunacovirus* in the S protein region. Moreover, an evolutionary tree built for the S1 region of Alpha-CoV revealed that the clustering trend of CoVs within the *Decacovirus* was closely related to their hosts. At the viral species level, apart from BatDeca_NL3r-CoVs, there was no differentiation in their clustering trends (Supplementary Fig. 3c).

The CoVs in *Minunacovirus* are predominantly identified in bats of the genus *Miniopterus* and are currently classified into two species, namely, *Miniopterus bat coronavirus 1* (MiBt1A) and *Miniopterus bat coronavirus HKU8* (MiHKU8). Phylogenetic analysis based on the ORF1ab revealed 33 newly identified CoVs possessing complete genomes within *Minunacovirus*, including 19 CoVs in MiBt1A and 14 CoVs in MiHKU8, all of which were primarily found in China (Fig. 5 and Supplementary Data 9). Notably, MiHKU8 related CoVs (MiHKU8r-CoVs) formed two evolutionary clades (A and B) with significant differences. The CoVs in HKU8-A were mostly from *Miniopterus Pusillus* (*M. Pusillus*) in Guangdong and Hongkong. While CoVs in HKU8-B were primarily identified from *M. schreibersii* and *M. Fuliginosus* in Yunnan, Guangdong, Fujian, Hubei, and Hainan provinces. Similarly, MiBt1A related CoVs (MiBt1Ar-CoVs) also had two significantly different clades (A and B). The former were largely detected in *M. schreibersii* from Yunnan, Guangdong, Anhui, Hubei, Jiangxi, Hongkong, and Sri Lanka, while the latter were mainly found in *M. pusillus* from Guangdong, Hainan, Hongkong, and Yunnan provinces.

Regarding the recombination analysis for the *Minunacovirus*, a total of 24 recombinant sequences were identified, present in both MiBt1A and MiHKU8 (Fig. 5a). We found that recombination events predominantly occur within MiBt1A or within MiHKU8, without involvement of inter-species recombination events (Supplementary Data 8). Moreover, by observing the phylogenetic tree constructed from ORF1a, ORF1b, and structural proteins, it was found that one evolutionary cluster in MiHKU8 diverged in the S protein region with other CoVs in *Minunacovirus*, showing closer phylogenetic relationship with WA3607 in *Decacovirus*. In addition, two sequences belonging to MiHKU8 were clustered in the N protein region into MiBt1A. Noteworthy is the fact that a virus strain identified from Civet (OM480510) was observed in the divergent evolutionary cluster in MiHKU8. From the phylogenetic tree constructed from the S1 region (Supplementary Fig. 3c), the amino acid sequence consistency of this virus strain with the virus strain identified in *M. fuliginosus* (KJ473798_Bat/MiFu_HB/CHN) in this region is as high as 80.39%.

Currently, four CoV species of *Nyctacovirus* are defined, namely *Nyctalus velutinus Alphacoronavirus SC-2013* (NySC13), *Alphacoronavirus WA2028* (WA2028), *Alphacoronavirus HKU33* (TyHKU33) and *Pipistrellus kuhlii CoV 3398* (PKBt3398). Whereas, taxonomic results derived from seven CRDs demonstrated the presence of two evolutionarily distinct lineages of unclassified species within *Nyctacovirus*, designated as BatNycta_NL6 and BatNycta_NL7 (Fig. 3a). The natural host of NySC13 was *Nyctalus velutinus* from Sichuan province. WA2028 related CoVs were found from the genus *Chalinolobus, Myotis*, and *Vespadelus* in Australia, and TyHKU33 related CoVs were traced back to *T. robustula* in China. CoVs pertaining to PKBt3398 were found in *Pipistrellus kuhlii* (*P. kuhlii*) in Italy. CoVs within BatNycta_NL6 showed extensive host diversity and wide regional distribution, while those within BatNycta_NL7 demonstrated a considerably narrow host range, confined exclusively to *P. abramus* in Guangdong, Guangxi, and Guizhou provinces. Nine newly identified CoVs (Complete Genome) in *Nyctacovirus*, which belonged to BatNycta_NL7 (*n* = 8) and TyHKU33 (*n* = 1) were based on the ORF1ab tree (Fig. 5 and Supplementary Data 9). CoVs belonging to BatNycta_NL7 were from *P. abramus* in Guangdong, Guangxi, and Guizhou provinces. The one CoV in TyHKU33 was from *Tylonycteris robustula (T. robustula)* in Guizhou province.

About the recombination analysis within the *Nyctacovirus*, 5 recombinant sequences were identified, all of which belonged to Bat-Nycta_NL7 (Fig. 5a). Apart from one recombination event involving BatNycta_NL7 and NvSC13, all other recombination events occurred within the BatNycta_NL7 species (Supplementary Data 8). Moreover, phylogenetic analyses based on ORF1a, ORF1b, and structural proteins revealed that the CoVs in *Nyctacovirus* demonstrated stability, with no significant shifts detected across different evolutionary clades (Supplementary Fig. 4).

The CoVs in *Rhinacovirus* are predominantly found in the genus *Rhinolophus* in China. To date, a single defined CoV species, *Rhinolophus bat coronavirus HKU2* (RhHKU2), has been identified. Based on the clustering in the evolutionary tree, CoVs in RhHKU2 have formed two distinct evolutionary clades (A and B). Notably, the SADS-CoV, responsible for the swine acute diarrhea syndrome observed in Guangdong province between 2016 and 2017, was categorized under HKU2-A[31]. According to the phylogenetic analysis based on the ORF1ab, 30 out of the 42 CoVs with complete genomes in this study were found to belong to HKU2-A, and 12 CoVs were grouped under HKU2-B (Fig. 5 and Supplementary Data 9). The CoVs in HKU2-A were predominantly derived from *R. affinis* and *R. sinicus* across Guangdong, Guangxi, Yunnan, and Hainan provinces. In addition, a small number of CoVs within HKU2r-A were detected in *R. ferrumequinum* and *M. laniger*. The CoVs within HKU2-B, however, were all traced back to *R. pusillus* in Zhejiang, Guangxi, and Yunnan provinces.

Regarding the recombination analysis for CoVs in *Rhinacovirus*, a total of 22 recombinant sequences were identified (Fig. 5a), and all recombination events occurred within the RhHKU2 species (Supplementary Data 8). However, phylogenetic analyses based on ORF1a, ORF1b, and structural proteins indicated no significant changes in the clustering trend of CoVs in *Rhinacovirus*. A singular deviation was noted in the S protein region, where CoVs coincided in the same evolutionary cluster with WA3607r-CoVs from *Decacovirus*, possibly suggesting a shared evolutionary trait or a cross-subgenus recombination event. Importantly, it should be mentioned that due to the lack of extant research findings on S1 or receptor-binding domain (RBD) in CoVs within *Rhinacovirus*, this investigation did not delve into such analysis (Supplementary Fig. 3c).

The CoVs in *Myotacovirus* are primarily discovered in the genus *Myotis*, which are broadly distributed across Asia and Europe. At present, only one species has been confirmed, namely *Myotis ricketti alphacoronavirus Sax-2011* (MaSaX11). Phylogenetic analysis anchored on ORF1ab disclosed that MaSaX11-related CoVs (MaSaX11r-CoVs) in *Myotacovirus* divided into two unique evolutionary clades (A and B) (Fig. 5 and Supplementary Data 9). 16 of 20 newly identified CoVs with complete genomes in this study from *M. ricketti* in Guangdong belonged to MaSaX11-A, and the rest of the four newly identified CoVs belonged to MaSaX11-B from *M. adversus* and *M. siligorensis* distributed in Guangxi, Jiangxi, and Hubei provinces.

Recombination analysis in *Myotacovirus* identified 9 recombination sequences within MaSaX11 (Fig. 5a and Supplementary Data 8). Phylogenetic analyses conducted on ORF1a, ORF1b, and structural proteins suggest that the clustering tendencies of CoVs in *Myotacovirus* show relatively minor variations across different evolutionary trees (Supplementary Fig. 4). In the S1 evolutionary tree (Supplementary Fig. 3c), different clusters of MaSaX11r-CoVs were found to be closely related to the host species.

The CoVs in *Pedacovirus* have been categorized into four species, namely *Porcine epidemic diarrhea virus* (PEDV), *Scotophilus bat coronavirus 512* (ScBt512), *Alphacoronavirus BT020* (MyBt020), and *Alphacoronavirus WA1087* (WA1087). However, the taxonomic results based on seven CRDs revealed the existence of two evolutionary lineages of as yet undetermined species in *Pedacovirus*, labelled as BatPeda_NL8 and BatPeda_NL9 (Fig. 3a). PEDV is recognized as a crucial pathogen causing pig epidemic diarrhea, which could induce acute diarrhea or vomiting, dehydration, and high mortality in neonatal piglets. Its widespread distribution in pig populations has been reported globally[32]. ScBt512 related CoVs (ScBt512r-CoVs) were primarily identified in the genus *Scotophilus* from Southern China and Southeast Asia, with a small number also recorded in Africa. WA1087 related CoVs (WA1087r-CoVs) have been detected in *Chalinolobus gouldii* in Australia. BatPeda_NL8 related CoVs were found in *Murina leucogaster* in China and *R. ferrumequinum* in South Korea. MyBt020 related CoVs were mainly found in the genus *Myotis*, which were distributed across Asia and Europe (Fig. 5 and Supplementary Data 9).

Phylogenetic analysis based on the ORF1ab showed that the 84 newly identified CoVs (Complete Genome) within *Pedacovirus* are linked to ScBt512 (*n* = 55) and MyBt020 (*n* = 29). Although ScBt512r-CoVs are primarily derived from *S. kuhlii*, they have evolved into two distinct evolutionary clades (A and B). These ScBt512r-CoVs are largely concentrated in Guangdong, Guangxi, and Hainan provinces. Notably, a CoV belonging to ScBt512-B has been detected in *S. heathii* from Yunnan province. MyBt020r-CoVs, on the other hand, are primarily distributed across Yunnan, Guangxi, Guangdong, HongKong, Jiangxi, and Hubei provinces, further clustering into six distinct evolutionary clades (A to F). CoVs in MyBt020-A and MyBt020-B were primarily detected in *M. ricketti*, while those in MyBt020-C were mainly identified in *M. siligorensis*. a CoV from MyBt020-D was discovered in *M. adversus* in Jiangxi province. The genetic distance among BatPeda_NL9-related CoVs is substantial, and they have been identified in various bat species of Myotis distributed in China and Korea. No CoVs belonging to WA1087, BatPeda_NL8, MyBt020-E and MyBt020-F were identified in this study.

In *Pedacovirus*, 60 recombinant sequences involving PEDV, BT020, BatPeda_NL9, and ScBt512 were identified (Fig. 5a), primarily concentrated within BT020 and ScBt512. It is noteworthy that the major recombination events occurred within the species, particularly within ScBt512 (Supplementary Data 8). Phylogenetic analyses conducted on ORF1a, ORF1b, and structural proteins revealed significant variations in clustering tendencies between sequences from ScBt512-A and ScBt512-B. Moreover, a strain (OQ175214) found in *M. ricketti* within BatPeda_NL9 clustered with BatAlpha_NL1r-CoVs in unclassified *Alphacoronavirus*. Notably, a distinct strain of Swine enteric CoV (SwineCoV, NC028806) that belongs to *Tegacovirus* exhibits significant recombination characteristics. In its genome structure, all regions apart from the S protein align with the transmissible gastroenteritis virus (TGV) in *Tegacovirus*, whereas the S protein region is highly analogous to PEDV in *Pedacovirus* (Supplementary Fig. 4). Furthermore, the phylogenetic tree constructed through S1 showed that recombination events within BT020 and ScBt512 primarily occur among hosts of the same species or genus, except for a few strains in PEDV and BatPeda_NL9 (Supplementary Fig. 3c).

Remarkably, four unclassified Alpha-CoVs have been identified within E. serotinus in Jiangxi province. The taxonomic results, drawn from seven CRDs, suggest that these newly discovered CoVs are categorized into BatAlpha_NL1 (Fig. 3a). BatAlpha_NL1r-CoVs were primarily found in the genus *Episticus*, exhibiting wide geographical distribution, and have been found in China, South Korea, and the United States. The examination of BatAlpha_NL1r-CoVs revealed four unique recombinant sequences, which presented profound recombination traits upon comprehensive phylogenetic analysis of ORF1a, ORF1b, and structural proteins (Fig. 5, Supplementary Fig. 4 and Supplementary Data 9). Distinguished primarily in the S protein region, these viral strains separate themselves from the rest of the BatAlpha_NL1r-CoVs contingent. Notably, they co-clustered with a CoV (OQ175214) from BatPeda_NL9 of *Pedacovirus*, which was originally identified in *M. ricketti* in Jiangxi province, China.

## Evolutionary, host range, distribution, and recombination characteristics of *Betacoronavirus* in bats

Presently, there are four recognized CoV species within *Merbecovirus*, including *Hedgehog CoV 1* (HeCoV1), *Middle East Respiratory Syndrome-related CoV* (MERS), *Pipistrellus bat CoV HKU5* (HKU5), and *Tylonycteris bat CoV HKU4* (HKU4). Taxonomic analysis based on seven CRDs confirmed the absence of undetermined species within *Merbecovirus* (Fig. 3a). MERS-CoV, categorized under the MERS species, has been identified as the causative agent of the Middle East Respiratory syndrome, exhibiting a case fatality rate of up to 30% in patients[33]. MERS-related CoVs (MERSr-CoVs) have a global distribution, spanning Asia, Europe, Africa, and South America, and are classified into two clades (A and B), MERS-A primarily originating from the Vespertilionidae family, and MERS-B primarily from the Camelidae family. In addition, HeCV1-related CoVs (HeCV1r-CoVs) are primarily distributed in Europe, with a minor population identified in China. Notably, HKU5r-CoVs and HKU4r-CoVs were all found from family Vespertilionidae in China. HKU5r- and HKU4r- CoVs, discovered within the family Vespertilionidae in China, have their related CoVs also identified in Bangladesh and Cambodia in Asia (Fig. 6 and Supplementary Data 9).

Phylogenetic analysis based on the ORF1ab showed in *Merbecovirus* a total of 64 CoVs (Complete Genome) were newly identified, involving HKU4 (*n* = 23), HKU5 (*n* = 33), and MERS-A (*n* = 8) CoVs (Fig. 6a). HKU4r-CoVs and HKU5r-CoVs exhibit significant host preferences. HKU4r-CoVs are largely found in *T. pachypus* across Guangxi and Guangdong provinces while HKU5r-CoVs are primarily detected in *P. abramus* in Guangdong, Guangxi, Jiangxi, Yunnan, and Zhejiang provinces. In contrast, MERSr-CoVs demonstrate a diverse range of hosts, encompassing nine bat genera, and are extensively distributed across Africa, Europe, and Asia. CoVs closely related to MERS-CoV have been identified from *Neoromicia capensis* and *M. ricketti* in South Africa. CoVs associated with MERS have been identified in five different bats of genus *Vespertilio* across Guangdong, Guangxi, Jiangxi, and Yunnan provinces, including *P. abramus*, *Io la*, *Eptesicus serotinus*, *T. pachypus*, *M. ricketti*, *H. larvatus*, and are categorized into two distinct evolutionary clades (A and B) (Fig. 6b and c).

Within the *Merbecovirus*, a recombination analysis identified 28 recombination sequences involving HeCV1, TyHKU4, PiHKU5, and MERS (Fig. 6). An important aspect to note is the notable diversity of recombination events within CoVs in *Merbecovirus*, occurring not only within viral species but also exhibiting several inter-species recombination events (Supplementary Data 8). Phylogenetic analyses conducted on ORF1a, ORF1b, and structural proteins revealed that MERSr-CoVs exhibited pronounced clustering patterns in the phylogenies of S and E proteins, while such significant trends were not observed in HeCV1r-, TyHKU4r-, and PiHKU5r- CoVs (Supplementary Fig. 5). Moreover, in the RBD region, MERSr-CoVs were found to cluster with HeCV1r-, TyHKU4r-, and PiHKU5r- CoVs, providing new insights into the evolutionary patterns of MERSr-CoVs (Supplementary Fig. 3d). The clustering trends of TyHKU4 and PiHKU5 were closely associated with their host species.

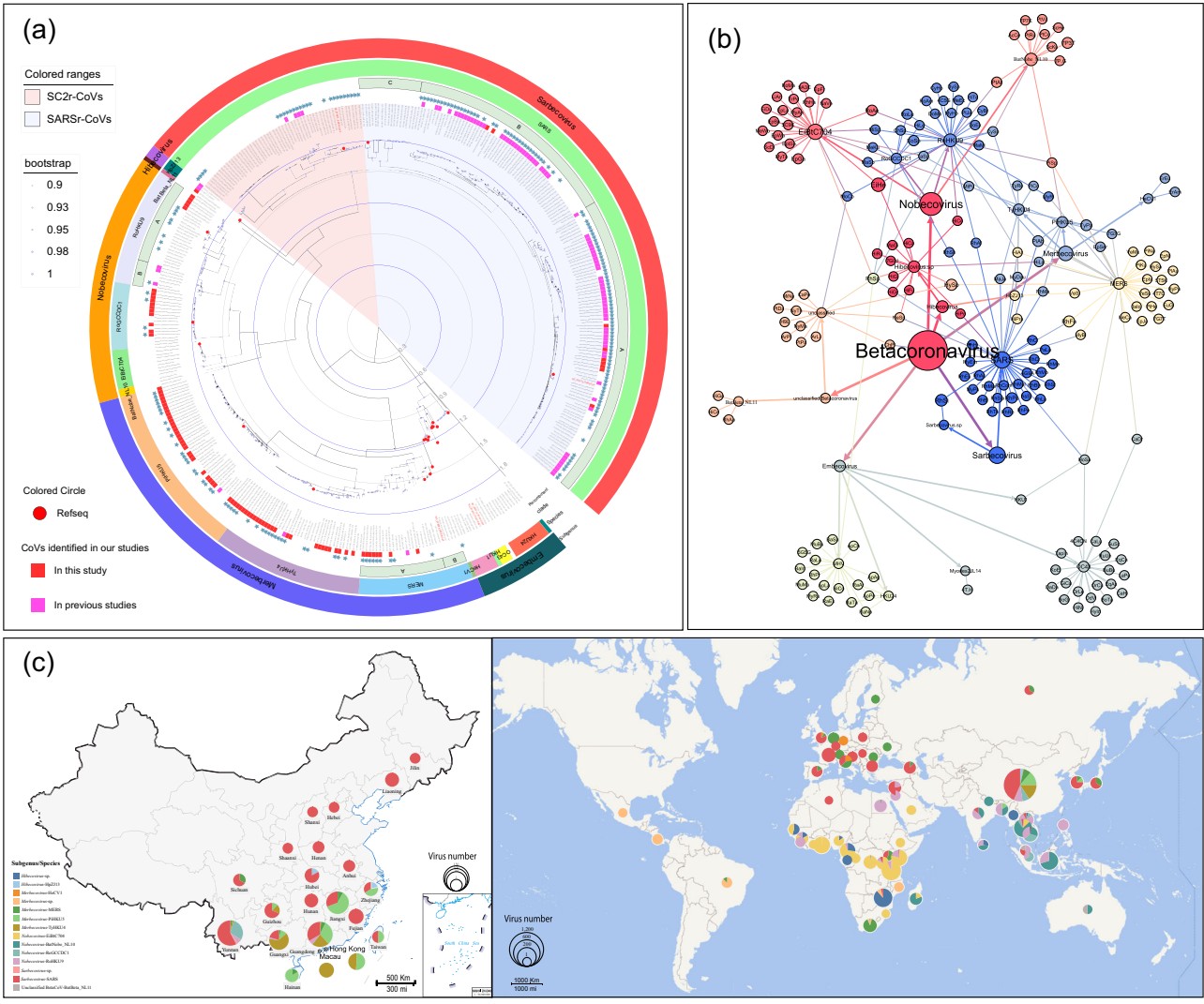

**Fig. 6 | Evolution, host range and distribution of coronaviruses in *Betacoronavirus*. a** The phylogenetic tree is based on the ORF1ab. The tree was constructed by the maximum likelihood method using appropriate models (GTR + I + G) based on iqtree. The viral species of each subgenus and their corresponding labels are present on the right of the phylogenetic tree. In addition, we have annotated the identified recombinant sequences, which were confirmed through RDP detection and manual corrections. These recombinant sequences have been marked with blue asterisks. **b** Network based on virus in *betacoronavirus* and the corresponding host. In the network, nodes represent the genus, subgenus, species, and hosts of viruses, with node size proportional to the number of viruses. We have employed abbreviations to show the viral species and hosts, with their corresponding full names available in Supplementary Data 2 and Data 6. Lines are

drawn between virus genus and its subgenus, subgenus and its species, and between each virus species and its corresponding hosts. To distinguish between the three levels of connections, the line widths decrease in the order from genus to subgenus to species. Running Community Detection named Modularity in the statistics page yields a Modularity Class value for each node, with node color ranking based on the Modularity class. The line colors change with the node colors. **c** The distribution and proportion in China and the world of each virus species in subgenus of *betacoronavirus*. The different colors in the pie chart represent different viral species. The size of each slice in the pie chart is proportional to the total number of viruses identified in that region. World map data were retrieved from Tianditu (https://www.tianditu.gov.cn/) and China map data were from https://doi.org/10.5281/zenodo.4167299.

The CoVs in *Nobecovirus* are principally derived from bats in the Pteropodidae family, segregated into three distinct species, *Rousettus bat coronavirus GCCDC1* (RoGCCDC1), *Rousettus bat coronavirus HKU9* (RoHKU9), and *Eidolon bat coronavirus C704* (EiBtC704). In contrast, taxonomic results based on seven CRDs denote an undetermined species in *Nobecovirus*, namely BatNobe_NL10, which is primarily found from *Pteropus poliocephalus* and *Pteropus rufus* in Southeast Asia, Oceania and Africa (Fig. 3a). RoGCCDC1 related CoVs (RoGCCDC1r-CoVs) and HKU9 related CoVs (HKU9r-CoVs) are primarily found in the genus *Eonycteris* and *Rousettus* from Yunnan Province of China, South Asia, and Southeast Asia. EiBtC704-related CoVs (EiBtC704r-CoVs), primarily located in the Pteropodidae family, have a wide geographical distribution across Africa, including Cameroon, Madagascar, and Congo. In this study, based on the ORF1ab, 12 of 28

newly identified CoVs (Complete Genome) belonged to GCCDC1r-CoVs, which were primarily found from *Eonycteris spelaea* and *R. leschenaulti* in Yunnan (Fig. 6 and Supplementary Data 9). A total of 16 CoVs were identified as HKU9r-CoVs, which were primarily found in *Rousettus* sp. and *Rousettus leschenaulti* from Yunnan, Guangxi, and Guangdong provinces.

Within the *Nobecovirus*, our recombination analysis identified 13 recombination sequences involving RoGCCDC1 and RoHKU9 (Fig. 6a), and recombination events primarily occur within RoGCCDC1 or within RoHKU9, with only one recombination event occurs between RoGCCDC1 and RoHKU9 (Supplementary Data 8). Phylogenetic analyses were conducted on ORF1a, ORF1b, and structural proteins, which displayed relatively consistent clustering patterns for CoVs in *Nobecovirus*. However, subtle variations were observed within the

phylogenies of the E protein and M protein of HKU9r-CoVs (Supplementary Fig. 5). Moreover, a trend of differentiation was observed within the evolution tree of RBD for HKU9r-CoVs (Supplementary Fig. 3d).

The CoVs in *Hibecovirus* are primarily associated with the family Hipposideridae, which has been characterized by a single species until now. This study has uncovered a CoV belonging to *Hibecovirus* from *H. pratti* in the Hubei province. This virus shares a 98% identity throughout the genome with Bat Hp-betacoronavirus/Zhejiang 2013, which is the only known CoV species of *Hibecovirus* to date (Fig. 6 and Supplementary Data 9). Notwithstanding, a significant volume of unclassified *Hibecovirus* was found in the genus *Hipposideros* from Africa, yet a lack of complete genome sequences precludes further taxonomic classification of these viruses (Fig. 3b).

The CoVs in *Sarbecovirus* are mainly from the genus *Rhinolophus*. These include SARS-CoV and SARS-CoV-2 that can cause serious respiratory disease in people. Among the 78 CoVs studied, 74 were identified as SARSr-CoVs, which evolved into three distinct clades (A, B and C) (Fig. 6a). These SARSr-CoVs were primarily specifically across a broad geographical range, with distinct clusters in South and Central China, Northeast China, and across Africa, Europe, and East Asia (Fig. 6 and Supplementary Data 9). Through our investigations, four strains within *R. pusillus* were identified, each with genome sequences demonstrating only 87.8%–90% nucleotide identity with the complete genome of SARS-CoV-2. Notably, these strains displayed apparent genomic recombination characteristics, earning them the previous classification as recombinant lineage (L-R)[29].

### Insights into the recombination process of SARS-CoV-2 and the zoonotic potential of two SARS-related coronaviruses

A comprehensive heatmap analysis of the genomic structure of known SC2r-CoVs was carried out to further probe their relationship with the L-R lineage (Fig. 7). This approach enabled the categorization of SC2r-CoVs into seven distinct groups, based on their phylogenetic relatedness to various regions of the SARS-CoV-2 genome. The detailed findings for each group, including their geographical distribution, nucleotide identity, and main host species, are shown in Fig. 7. The genomic structure heatmap uncovered those strains from Groups 3, 4, and 5 collectively encompass the gene elements constituting SARS-CoV-2, suggesting that SARS-CoV-2 likely originated from multiple recombination events involving strains from these groups. Group 2 strains could be interpreted as recombinants of Group 3 strains distributed mainly in China and Group 4 strains from the Indochina Peninsula, with ORF1a and ORF1b potentially contributed by the respective groups. Subsequently, Group 1 strains may have evolved from Group 2 by acquiring S protein fragments from Group 5.

In order to validate this hypothesis, a Genetic Algorithm for Recombination Detection (GARD) was implemented, which identified seven recombination breakpoints in the evolutionary trajectory of SC2r-CoVs (Fig. 7). These breakpoints, corresponding to various regions in the ORF1a (12,764), ORF1b (21,401), and the S protein (21,401 and 24,426) of SARS-CoV-2, align well with our projected recombination locations. It is noteworthy that the strains such as BtSY2 (OP963576) from Group 3 and RShSTT182 (EPIISL852604) from Group 4, indicate potential ongoing complex recombination among SC2r-CoVs, which could potentially give rise to novel CoV genotypes. In a further enhancement of our analysis, an evolutionary dendrogram, curated from the RBD of the Beta-CoV, and the aligned and collated RBDs of the *Sarbecovirus*, imparted insightful findings. Strains bearing a close kinship with the RBD sequence of SARS-CoV-2 emerged from a diverse array of hosts. These hosts spanned from the *Malayan pangolin* (MT121216_MP789) to varying species of bats such as *R. marshalli* (BtSY2), *R. pusillus* (MZ937001_BANAL103), and *R. malayanus* (MZ937000_BANAL52), signifying the broad host range of these viruses.

In contrast to other SC2r-CoVs, SARS-CoV-2 features a furin site of unknown origin nestled between the S1 and S2 subunits of its S protein. A comprehensive survey of the presence of furin sites across all genome datasets (Dataset 1) was conducted to explore possible sources of this furin site (Supplementary Fig. 7). Despite its rarity in *Sarbecovirus*, furin sites were discovered in the S protein of various coronavirus subgenera in Alpha- and Beta-CoV. Furin sites were detected in the S protein of *Amalacovirus*, *Setracovirus*, *Minunacovirus*, *Hibecovirus*, *Nobecovirus*, *Embecovirus*, *Merbecovirus*, despite being exceptionally rare in *Sarbecovirus*. It is worth noting that the highest prevalence of furin sites was recorded in *Embecovirus*, *Merbecovirus*, and *Hibecovirus*, primarily occupying the space between S1 and S2. Contrarily, in other subgenera, the furin sites were primarily confined within the S2 subunit (Supplementary Fig. 3d and Supplementary Data 10).

To elucidate the potential infectivity of the identified CoVs in *Sarbecovirus*, an in-depth structural analysis of their RBDs relative to the hACE2 receptor, a key determinant for cellular entry, was conducted. A comparative alignment of non-redundant RBDs across CoVs in *Sarbecovirus*, underscoring the contacting residues to SARS-CoV and SARS-CoV-2 interaction with hACE2, allowed for the visual inspection of the amino acid variability at these contact points. This framework informed the selection of the RBDs of SARSr-CoV YN2020B (OK017852) and SC2r-CoV HN2021A (OK017803) for thorough investigation. YN2020B, due to its highest nucleotide similarity (95.8%) to SARS-CoV, and HN2021A, the representative strain we identified to be associated with SARS-CoV-2, were chosen for deeper analysis. YN2020B's RBD displayed notable amino acid identity and structural congruence with SARS-CoV, suggesting potential hACE2 receptor affinity despite five amino acid variations. In contrast, HN2021A's RBD, while presenting a 65.92% amino acid identity with SARS-CoV-2's RBD, revealed significant structural divergences and variations at key contact residues, implying possible infectivity variations (Supplementary Fig. 8). These preliminary insights, though speculative, highlight the need for further experimental validation to definitively confirm the infectivity and host range of these CoVs in *Sarbecovirus*.

### Coevolutionary and cross-species transmission of bat CoVs

Based on the reconstruction of ancestral hosts, we found that the evolutionary origins of Alpha-CoV can be traced back to the genus *Myotis* of family Vespertilionidae. Within bat-associated Alpha-CoVs, the *Rhinacovirus* emerged first, predominantly originating from bats belonging to the genus *Rhinolophus* (Fig. 8a). As for Beta-CoV, its evolutionary origins were associated with genus *Hipposideros* of family Hipposideridae and genus *Rattus*. Among bat-associated Beta-CoVs, the *Nobecovirus* first formed, originating primarily from bat species within the genus *Rousettus* (Fig. 8b). The MCC tree revealed a discernible coevolutionary relationship between the hosts and CoVs. While most branches exhibit a clear association with a single-host family or genus, there are still some CoVs that show strong associations with other host families or genera, suggesting the occurrence of frequent cross-species transmission events.

Hence, at the genus level of host classification, we identified 30 Bayesian-supported intra-species host switches for Alpha-CoVs and 29 for Beta-CoVs (Fig. 8c and d). Alpha-CoV, there was evidence of frequent host switches between hosts in family Vespertilionidae and those in other families, supported by high Bayesian factors, particularly for genera *Tylonycteris*, *Nyctalus*, and *Chalinolobus*. Several genera, such as *Hipposideros*, *Megaderma*, *Rhinolophus*, primarily functioned as donors (Supplementary Table 1). As for Beta-CoV, Vespertilionidae emerged as the dominant donor, with significant support for host switches to genera *Cynopterus*, *Eonycteris*, and *Hypsugo*, all evidenced by Bayesian factors surpassing 100. The genus *Eptesicus* of Vespertilionidae showed evidence of a switch to genera *Pteropus* and *Aselliscus*, with supporting Bayesian factors over 100. Pteropodidae bats primarily acted as receivers, with genus *Micropteropus* exclusively

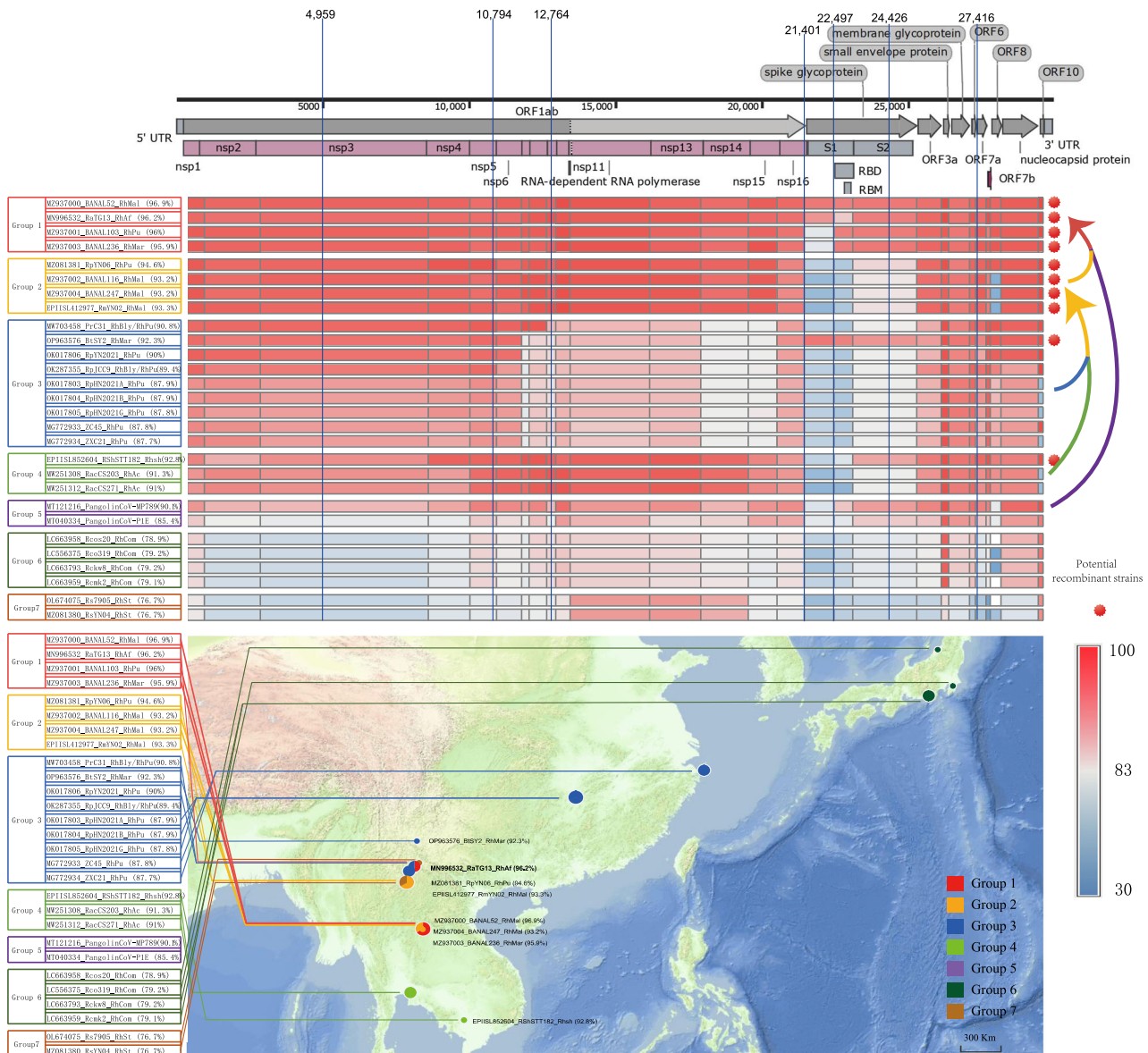

**Fig. 7 | Overview of SC2r-CoV genomic structure, recombination events, and geographical distribution.** Genomic structure heatmap of SC2r-CoVs strains categorized into seven distinct groups based on their phylogenetic relatedness to various regions of the SARS-CoV-2 genome. Each row represents a SC2r-CoV strain, while each column represents a segment of the SARS-CoV-2 genome. The color gradient represents the degree of nucleotide identity, with lighter colors indicating higher identity. Breakpoint analysis for the detection of recombination events. Within the evolutionary trajectory of SC2r-CoVs, seven distinct recombination breakpoints were identified. The breakpoints correspond to positions 4959, 10,794, 12,764, 21,401, 22,497, 24,426 and 27,416 on the genome of SARS-CoV. Each breakpoint is marked by a blue vertical line and corresponds to specific regions within ORF1a, ORF1ab, and the S protein of the SC2r-CoV genome. Geographical distribution map of SC2r-CoVs. Different colors indicate the seven groups of SC2r-CoVs, with each dot representing a specific SC2r-CoV strain. The geographical distribution provides a visual representation of the prevalence and dispersal of SC2r-CoV strains across various regions. Map data were retrieved from Tianditu (https://www.tianditu.gov.cn/).

receiving switches from other genera, backed by Bayesian factors all exceeding 100. Similarly, genera *Myonycteris*, *Macroglossus*, and *Eonycteris* functioned exclusively as receivers (Supplementary Table 2).

## Discussion

Since the onset of the 21st century, epidemics or pandemics due to highly pathogenic CoVs such as SARS-CoV, MERS-CoV, SADS-CoV, and SARS-CoV-2 have posed significant threats to human or animal health, resulting in substantial disruptions to daily life and economic development[1–3,31]. Importantly, all EIDs precipitated by these CoVs have been traced back to their highly homologous counterparts present in

bats[3,4,34]. As research progresses in this field, it has been discovered that bats host a diverse range of Alpha-CoVs and Beta-CoVs[10]. Expanding our understanding of the host and CoV diversity and distribution may yield valuable information to enhance targeted surveillance and spillover prevention programs.

Here, a large-scale and in-depth investigation of bat CoVs was conducted across the southern regions of China. These regions included provinces such as Yunnan, Guangxi, and Guizhou, which are known for their high prevalence of bat CoVs (Supplementary Data 6). In addition, provinces like Hubei and Guangdong, where CoV-related diseases were first reported[1,3], and provinces such as Hunan[29] and Zhejiang[35], where novel CoV clades were discovered,

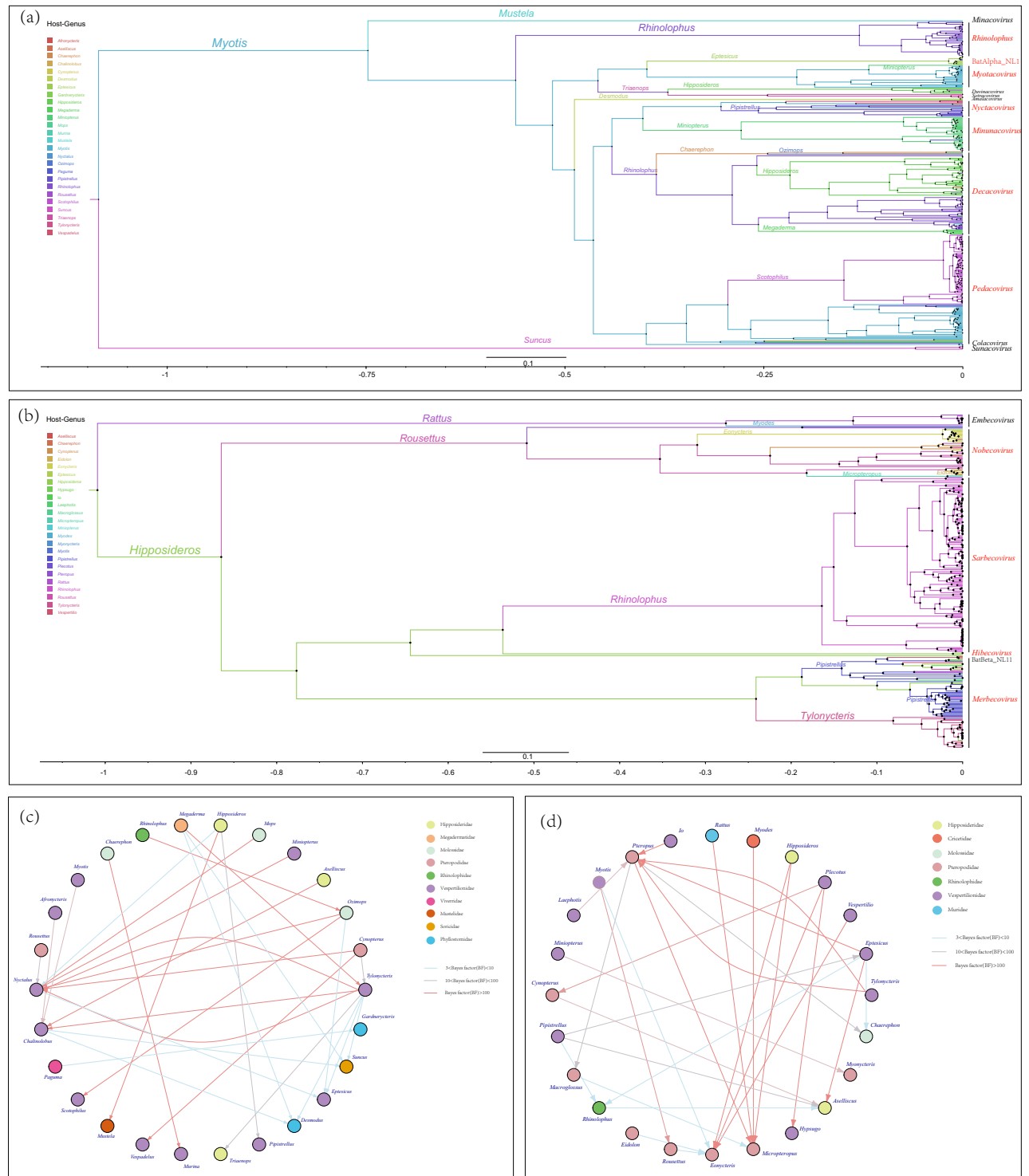

**Fig. 8 | Ancestral host reconstructions and inter-genus host switches. a** and **b** Represent the maximum clade credibility (MCC) trees constructed separately for alphaCoV and betaCoV using full RDRP sequences and the corresponding host genus as discrete character states. The color of branches in the trees is consistent with the inferred ancestral host. Branch lengths are scaled based on relative time units, where the clock rate is set to 1.0. The corresponding virus subgenus is labeled on the right side of the trees. Host switches between genus for (**c**) Alpha-CoVs and (**d**) Beta-CoVs are shown only host switches supported by a strong Bayes factor (BF) > 3. The blue lines indicate host switches with a Bayesian factor between 3 and 10, gray lines represent host switches with a Bayesian factor between 10 and 100, and pink lines represent host switches with a Bayesian factor >100. The arrows indicate the direction of the host switch, and the nodes that belong to the same family have the same color.

were also included in the study. Through the identification of new bat CoVs, a comprehensive analysis was conducted on the overall characteristics of bat CoVs in China, in terms of phylogenetic evolution, genetic recombination, and cross-species transmission. This has led to further refinement in the taxonomies of bat Alpha-

and Beta-CoVs, and contributed to a basic understanding of bat CoVs in China.

Existing research into bat CoVs suggests that bats predominantly harbor approximately 16 classified CoV subgenera, while bats found in China are associated with about 10 such subgenera[9]. Beyond the ten

classified CoV subgenera linked to bats found in China[9], this study has further identified an additional unclassified subgenus of Alpha-CoV. It is important to note that there may still be other classified CoV subgenera borne by bats in China that remain undiscovered. Nonetheless, this data elucidates the primary types of CoV subgenera borne by bats within China. Conforming to the classification standards provided by the ICTV, multiple CoV species within the known CoV subgenera were identified and named based on their phylogenetic and host characteristics. Interestingly, a substantial number of novel Alpha-CoVs and Beta-CoVs were identified, with a notable predominance of the Alpha-CoV, including *Decacovirus*, *Nyctacovirus*, and *Pedacovirus*. Intriguingly, no novel species of Beta-CoV were discovered in China. This observed inclination could be due to a greater diversity of CoVs within Alpha-CoV, or it may be influenced by potential sampling biases.

A detailed analysis into the host characteristics of CoVs within different subgenera showed that *Rhinacovirus*, *Myotacovirus*, and *Sarbecovirus* are predominantly hosted by a single bat genus, either *Rhinolophus* or *Myotis*. Other CoV subgenera found hosts within two or more bat genera, forming distinct evolutionary clades. Remarkably, bats within genera *Rhinolophus*, *Myotis*, *Hipposideros*, and *Pipistrellus* were carriers of multiple different CoV subgenera (Supplementary Data 2, Data 6 and Data 9). Of significant note, bats of the genus *Rhinolophus* were found to be the primary natural hosts of SARS-CoV, SADS-CoV, and SARS-CoV-2[9,31,36]. The mechanisms and routes by which CoVs harbored by bats of the genus *Rhinolophus* frequently spillover to human society remain unclear, thereby underscoring the need for heightened vigilance concerning CoVs carried by bats of this genus.

Our study has unveiled intriguing insights into the recombination process of SARS-CoV-2. The lineage L-R (Group 3) of *R. pusillus*, as pointed out by a previous study[29], is an evolutionary intermediate between SARS-CoV and SARS-CoV-2. Yet, deciphering the precise recombinant relationship between these entities necessitates further dedicated research efforts. One key finding in our study is the identification of three unique characteristics of the CoVs in Group 3: (1) a widespread distribution across China, from the Southwest to the Eastern regions, showing a progressive decrease in sequence identity with SARS-CoV-2 as geographical distance from the Yunnan province's border area increases; (2) substantial genetic diversity correlating with different distribution regions; (3) a specific host, *R. pusillus*. These characteristics strengthens the hypothesis of their enduring circulation within *R. pusillus* over an extended period. The realization of their extensive distribution and noticeable genetic diversity across various regions of China is in stark contrast with the limited distribution and diverse hosts of CoVs closely related to SARS-CoV-2 in Group 1 (such as RaTG13 and BANAL-20-52)[3,37,38]. However, in comparison to CoVs in Group 3, SARS-CoV-2 displays unique sequence identities and sources in certain regions that warrant further exploration.

An in-depth heatmap analysis, alongside the implementation of GARD, unraveled complex recombination events in the evolutionary trajectory of SC2r-CoVs. Interestingly, these events involve strains from Groups 3, 4, and 5, which collectively encompass the gene elements constituting SARS-CoV-2. Further, Group 2 strains appear to be recombinants of Group 3 strains distributed mainly in China and Group 4 strains from the Indochina Peninsula, with ORF1a and ORF1b potentially contributed by the respective groups. Subsequently, Group 1 strains may have evolved from Group 2 by acquiring S protein fragments from Group 5. This finding highlights the likelihood of SARS-CoV-2's origin in multiple recombination events, resonating with the theory of SARS-CoV's origin[39]. However, this raises a compelling question regarding the underlying mechanisms and driving factors of such complex recombination events within these specific clades of SC2r-CoVs.

In this study, CoVs closely related to SARS-CoV and SADS-CoV were detected[9], with the nucleotide sequence identity in the whole genome at 95.80% and 97.85%, respectively. Notably, SARSr-CoVs were predominantly found in *R. sinicus* from Yunnan province, while SADSr-CoVs were distributed across Yunnan, Guangxi, Guangdong, and Hainan provinces, with *R. affinis* as the main host. Bats from Yunnan province have been identified to carry SARSr-CoVs capable of directly utilizing human angiotensin-converting enzyme 2(hACE2)[40–42]. Furthermore, we conducted a comparative analysis between our identified sequences and SADS-CoV, focusing on the N-terminal domain (NTD) and C-terminal domain (CTD) regions of the S protein. The results revealed a high amino acid sequence identity of 99.22% (NTD) and 98.02% (CTD) between our identified sequences (OQ175189, OQ175199, and OQ175202) and SADS-CoV in these specific regions. This finding suggests that SARSr-CoVs and SADSr-CoVs present in the natural environment have the potential to directly transmit to humans or swine, thereby posing a risk of causing diseases. A detailed examination of the RBD of CoVs in *Sarbecovirus* carried by bats and pangolins has revealed a close phylogenetic relationship in RBD region with SARS-CoV and SARS-CoV-2. This evidence suggests that bat CoVs, including SARS-CoV, SARS-CoV-2, and SADS-CoV, which can infect humans or livestock, may have highly homologous sequences in the RBD or S1 region that are present in the natural environment. This finding highlights the need for further investigation and attention. Furthermore, our study has identified numerous recombination events among the 11 subgenera of CoVs. This indicates that frequent recombination events may occur, resulting in the rearrangement of RBDs or S1 regions in naturally circulating CoVs that have the potential to infect humans or livestock, leading to the emergence of new CoVs. These newly generated CoVs could potentially be transmitted to human societies through new bat hosts or other animal hosts. Our results suggest that SARS-CoV-2 may have undergone multiple recombination events in recent evolutionary time. This may indicate that the propensity of CoVs in *Sarbecovirus* to recombine is critical to their ability to emerge as human pathogens, and therefore that further surveillance of these viruses in wildlife should also consider recombination among strains as a risk factor for public health.

MERS-CoV, a member of the MERSr-CoV species in *Merbecovirus*, is characterized by a broad host range. In addition to bats of the genera *Pipistrellus*, *Eptesicus*, *Vespertilio*, and *Myotis*, closely related MERSr-CoVs have also been identified in Dromedary camels[43]. These bat MERSr-CoVs are distributed across Africa, Europe, and Asia. However, in the S protein region, bat MERSr-CoVs from Russia and South Africa were clustered into the HeCoV1 species, while bat MERSr-CoVs from China were grouped into the HKU5r-CoV species (Supplementary Data 2, Data 6 and Data 9). As seen in previous CoVs capable of cross-species transmission, bat MERSr-CoVs exhibit high host diversity. Nevertheless, the sequence identity of the S protein from related hosts is low with MERS-CoV, possibly accounting for various intermediate hosts of MERS-CoV, inadequate adaptation to humans and its sustained presence in the human population, which warrants further investigation. The role of hedgehogs in the origin of MERS-CoV remains uncertain, and it is unclear whether HeCoVs, like pangolin CoVs, have contributed sequence fragments to the formation of MERS-CoV. HKU5r-CoVs and HKU4r-CoVs were found in the bats of the genus *Pipistrellus* and genus *Tylonycteris*, respectively, in China. Given their single-host species and high sequence identity, they are initially assessed as having a relatively low risk of cross-species transmission to trigger an epidemic in the near term.

Many CoVs belonging to *Pedacovirus* were identified from the bats of the genera *Scotophilus* and *Myotis*. Although these CoVs had low sequence identity in the whole genome with PEDV, a very chaotic recombination lineage was formed among many CoVs from *S. kuhlii*. An interesting phenomenon is the group of unclassified Alpha-CoVs, also clustered into MyBt020r-CoVs of *Pedacovirus* in the S protein region. Although the unclassified Alpha-CoVs and a CoV (OQ175214) of MyBt020r-CoVs were found in different bat species, they were both found in Jiangxi province, China. This further proves that CoVs carried

by different animal species with spatial intersection can obtain new gene fragments in the process of jumping between different species. In general, CoVs in *Pedacovirus* showed a characteristic of easier recombination, which may increase the risk of cross-species transmission of this subgenus.

The CoVs in *Decacoviruses* exhibits a diversity of hosts, prominently including bats of the genus *Rhinolophus*. This subgenus has undergone a comprehensive reclassification, leading to the proposition of six novel species. Noteworthy is the fact that CoVs in *Decacovirus* showcase a wide range of host characteristics and an array of recombination patterns within the structural protein region. Moreover, regions in the genome structures of WA3607 and BatDeca_NL2, were clustered into *Minunacovirus* and *Rhinacovirus*, respectively. This observation, despite lacking direct evidence of recombination, intimates an intricate evolutionary relationship between these subgenera. Within *Minunacovirus*, the predominant hosts are bats belonging to the genus *Miniopterus*, accompanied by the discovery of a plethora of recombinant strains. It is intriguing to note that civets were reported to be carriers of HKU8r-CoVs. This warrants a deeper investigation to understand the mechanisms and pathways facilitating frequent spillover of CoVs from bats to civets. In *Nyctacovirus*, CoVs are harbored by bats of the genera *Nyctalus* and *Pipistrellus*, forming distinct evolutionary lineages reflective of their diverse hosts. Despite a broad geographic distribution, their identification in China has been relatively scarce. CoVs in *Myotacovirus*, *Nobecovirus*, and *Hibecovirus* manifest a narrower host range, spanning bats of genus *Myotis*, family Pteropodidae, and genus *Hipposideros*. Within the realm of current research, CoVs identified within *Hibecovirus* were relatively sparse. Intriguingly, these limited CoVs present a unique ORF, positioned between the ORF1ab and S protein, bearing a resemblance to the S protein yet in a more abbreviated form. The biological relevance of this unique ORF remains an enigma, urging further investigation.

In conclusion, this study has presented substantial new data on the occurrence of bat-CoVs, including many novel whole-genome sequences. Our results provide reference data for a deeper understanding of evolutionary relationships among a group of viruses that has caused disease outbreaks in people and livestock. We have analyzed patterns of recombination, and host-virus relationships that may be able to assist in targeting surveillance and identifying key foci for spillover risk.

## Methods

### Ethics approval and consent to participate

Animals were treated according to the guidelines of the Regulations for the Administration of Laboratory Animals (Decree No. 2 of the State Science and Technology Commission of the People's Republic of China, 1988). Sampling procedures were approved by the Ethics Committee of the Institute of Pathogen Biology, Chinese Academy of Medical Sciences & Peking Union Medical College (Approval number: IPB EC20100415).

### Sample collection

Bats were captured in their natural habitats, which included karstic caves, forests, woods, or abandoned buildings, employing a combination of hand nets (for catching bats in caves or buildings), mist nets (for sealing the cave or building exits), and harp traps (for trapping bats in forests or woods)[29]. Following capture, each bat was stored in a cotton bag for swab collection. Pharyngeal and anal swabs were collected from live bats, with samples immediately immersed in virus sampling tubes (Yocon, Beijing, China) containing maintenance medium. These were initially stored in a portable cooler at −20 °C or in liquid nitrogen for temporary preservation during fieldwork. Subsequently, samples were transferred to our laboratory and stored at −80 °C for long-term preservation. Species identification of the bats

captured was initially conducted in the field based on morphological characteristics by experienced field biologists. To further ensure the accuracy of species identification, we conducted DNA barcoding analysis of the mitochondrial cytochrome b gene[44]. This analysis was performed on a randomly selected subset of bat samples, including a total of 69 patagium samples from bats. The results of DNA barcoding were found to be consistent with the morphological species identification performed by the experienced field biologists. The obtained barcoding data have been deposited in GenBank (ON640659-ON640727).

### Library construction and next generation sequencing

To increase the efficiency of our investigation and to mitigate the burden of library preparation and sequencing, we adopted a pooling approach for samples originating from the same bat species and collected from identical sites. Specifically, 1 ml aliquots of maintenance medium from each individual sample were merged into a single container, thus creating a composite sample. The number of samples amalgamated into a single composite varied, encompassing a minimum of 1, a maximum of 49, an average of 23.1 and an standard deviation of 9.66 (Supplementary Fig. 1). The rationale behind our pooling strategy was primarily the bat species and collection site, while the time of collection was also considered. Generally, we pooled ~30 samples together to form a pool, with the number of samples in a pool typically not exceeding 50. Our approach involved the following considerations: (1) Same Species at the Same Collection Site: We prioritized pooling samples of the same species collected from the same sampling site into a single pool. (2) Insufficient Sample Quantity: In cases where the quantity of a specific species at a collection site was insufficient for a single pool, we considered pooling samples of the same species from neighboring counties or cities. In addition, for a small number of samples in a province, we treated each sample as a separate pool and performed individual sequencing to ensure the sensitivity of detection. (3) Excessive Sample Quantity: If the quantity of a specific species at a collection site exceeded the capacity (more than 50) of a single pool, we divided the samples into multiple pools based on time of collection (e.g., yearly or monthly).

Following the pooling of samples, we utilized a virus-particle-protected nucleic acid purification method to process these combined samples[45]. The samples were first homogenized in virus maintenance medium. The virus maintenance medium was used to homogenize the samples. The samples were filtered through a 0.45 µm polyvinylidene difluoride filter (Millipore, Germany), then were centrifuged at $150,000 \times g$ for 3 h. A cocktail of DNase and RNase enzymes were used to digest the pellet. After extracting viral RNA using a QIAmp MinElute Virus Spin Kit (Qiagen, USA), we used the primer K-8N and a Superscript IV system (Invitrogen, USA) to synthesize first-strand viral cDNA. By Klenow fragment (NEB, USA), the cDNA was converted into dsDNA. Primer K was used to perform sequence-independent PCR amplification. Then magnetic beads (Beckman Coulter, USA) were used to purify the PCR products which were from 300 to 2000 bps. Then the Nextera® XT DNA Sample Preparation Kit (Illumina, USA) was used to construct libraries and an Illumina HiSeq X Ten sequencer was applied for a paired-end read of 150 bp. The previous criteria described were used for filtering the sequence reads[46].

### Taxonomic assignment and contigs assembly

The sequence reads were aligned to sequences in the NCBI non-redundant nucleotide database (NT) and non-redundant protein database (NR) using BLASTx.

MEGAN6[47] was applied to parse and extract the taxonomies of the aligned reads with the best BLAST scores (E score < 10 − 5). We used megahit v1.2.9[48] with default parameters to assemble coronavirus which had been extracted and assembled contigs as reference sequences when doing PCR screening and sequencing.

## PCR screening and genome sequencing

In this study, a total of 372 pooled samples were initially subjected to PCR screening using conserved primer pairs (F: 5′-GGTTGGGAC-TATCCTAAGTGTGA -3′; R: 5′-CCATCATCAGATAGAATCATCATA-3′)[49–51]. Out of these pools, 199 tested positive for CoVs. It is important to acknowledge that the sample pooling approach has inherent limitations. To address this, we conducted additional screening of individual samples to confirm the presence of CoVs. Following this, the individual samples corresponding to the positive pools were also screened, leading to the identification of 1141 positive individual samples. From these, we successfully obtained 399 full-length CoV genomes. The selection of these CoVs for whole-genome sequencing was based on a set of criteria. Firstly, we constructed a phylogenetic tree using the partial RdRp sequences obtained from our screening. Sequences that formed distinct evolutionary branches were prioritized for whole-genome amplification. Furthermore, for sequences located on the same evolutionary branch, one representative strain was selected for whole-genome amplification. Nested specific sequence primers, designed based on assembled contigs of coronavirus and their closest reference sequences identified by Blast from Genbank, were utilized to amplify the partial genomes. The PCR products were then sequenced using an ABI3500 DNA analyzer (Applied Biosystems, USA) for Sanger sequencing. PCR products with low concentration or generating heterogeneity in the sequencing chromatograms were cloned into pMD-18T Vector (Takara) for sequencing.

To assess the prevalence of bat CoVs across various bat species and geographical regions in China, we analyzed the screening results obtained from bat samples. We visualized these outcomes using heatmaps, which were generated with the pheatmap (version 1.0.12) and vegan (version 2.6.4) R packages in R version 4.2.3, drawing from the coronavirus positivity rates detailed in Supplementary Data 3.

## Genome assembly and annotation

In this study, we employed Geneious Prime software (version 2022.2.2) to assemble the viral genomes from PCR products. For each sample, the Sanger sequencing reads were imported into Geneious Prime and subjected to a de novo assembly process. Post assembly, these complete viral genomes were annotated using the 'Annotate & Predict' feature in Geneious Prime. Specifically, we utilized the 'Annotate from Database' function, which enabled us to compare our assembled sequences to a reference database of sequences. This database comprised reference sequences curated from the RefSeq database available on the National Center for Biotechnology Information (NCBI) platform. This process facilitated the prediction and annotation of coding sequences, and the identification of genomic features within the assembled viral genomes. The R package ggplot2 (version 3.4.2) was utilized to generate box plots illustrating the full genome lengths of the identified subgenera.

## Sequence data

To augment the comprehensiveness of our study, we incorporated CoV sequences from GenBank, supplemented by a selection from the China National Genomics Data Center, and GISAID into our dataset. This amalgamation facilitated a more complete overview of the CoV genomic landscape, thereby enriching our understanding of coronavirus evolution and host distribution. Our data collection protocol encompassed pivotal parameters such as Accession numbers, sequence source, Organism, Completeness, Length (nt), Sequence label, Host-Taxonomy, Host-Family, Host-Genus, Host-Species, Virus-Genus, Virus-Subgenus, Virus-Species, location, and Collection_date, etc. In total, we collected 5272 CoV sequences, 5181 of which originated from bats. To ensure the accuracy of the meta-information, we made corrections to the taxonomic information of unannotated CoV sequences based on their phylogenetic placement.

The final datasets comprised 1456 sequences generated for this study, and 5272 sequences obtained from GenBank, the China National Genomics Data Center, and GISAID. The Accession numbers for these sequences are detailed in Supplementary Data 2 and Data 6

## Phylogenetic and sequence identity

We first aligned sequences using MAFFT v7.475[52], then used iqtree[53] to find the optimal model for building the phylogenetic tree or FastTree[54] to construct the phylogenetic tree. Based on the optimal model, we constructed a the phylogenetic tree with 1000 bootstrap replicates, and then annotated the tree using Interactive Tree Of Life (itol)[55] by adding label, subgenus, species, and other information.

For the construction of the species cytochrome b (cytb) phylogenetic tree, we downloaded the cytb sequences of 54 confirmed species out of the 58 species involved in the sampling. Since *Nyctalus velutinus* and *Pipistrellus tenuis* do not have cytb sequences on NCBI, we constructed the phylogenetic tree using cytb sequences of the other 52 species. We used Geneious Prime software (version 2022.2.2) to annotate the sequences in the data sequences and extracted the seven conserved regions, ADRP, nsp5 (3CLpro), nsp12 (RdRp), nsp13 (Hel), nsp14 (ExoN), nsp15 (NendoU) and nsp16 (O-MT) of coronavirus. Based on these seven conserved regions, we constructed a taxonomic phylogenetic tree according to the standards of ICTV and used MegAlign (DNA Star package Lasergene v.7.0.1) to align the concatenated sequences of the seven conserved regions with the refseq of each subgenus. Based on the annotation information of the sequences, we constructed phylogenetic trees of *Alphacoronavirus* and *Betacoronavirus* based on the extracted 1ab gene, and further annotated them using Interactive Tree Of Life (itol)[55]. Furthermore, we acquired the world map data from Tianditu (https://www.tianditu.gov.cn/) and the China map data from 10.5281/zenodo.4167299, subsequently annotating the proportions of different virus species at their corresponding locations.

## Genome analysis

For the collected sequences of subgenera within *Alphacoronavirus* and *Betacoronavirus* that have been submitted to the NCBI database and the sequences identified in this study, we predict furin sites and scoring using the ProP server (v1.0)[56] with default settings. The prediction scores ranged from 0 to 1, where higher scores indicated higher prediction accuracy. We further refined the predicted furin sites based on two criteria. Firstly, furin sites with prediction scores below 0.5 were not considered the real sites. Secondly, furin sites located outside the S protein region were excluded from the analysis. Ultimately, a list of confirmed furin sites was compiled, along with sequences from the *Sarbecovirus*, a phylogenetic tree based on the amino acid sequences of the S protein was constructed, and it was annotated using Interactive Tree Of Life (itol)[55].

To analyze the RBD region within the *Sarbecovirus* subgenus, we merged sequences that exhibited 100% identity in the RBD region and annotated the corresponding host information in the sequence names. The sequences from the *Sarbecovirus* subgenus were aligned using MAFFT v7.475[52], and the RBD structure was annotated using Geneious Prime software (version 2022.2.2) for subsequent analysis.

Using SnapGene 5.3.1 software (http://www.snapgene.com), we generated genomic structure diagrams and breakpoint maps corresponding to the GB files. Manual modifications were made to provide specific annotations for the genomic structure.

## Network graph

Based on the data sequence section described above, we used RDRP virus sequences and constructed virus-host networks using Gephi[57]. We divided the viruses into *Alphacoronavirus* and *Betacoronavirus* to construct the network between viruses and hosts. We connected each genus to its corresponding subgenera, each subgenus to its

corresponding species, and each species to its corresponding host. We used Force Atlas for layout, adjusted the repulsion strength to 1000, and selected Adjust by Size to make the layout more coherent. For the node size setting, we used the Ranking module and selected the size based on the weight, with a minimum size of 10 and a maximum size of 40. For the node color setting, we first used the Modularity from the Community Detection in the statistical part to calculate the corresponding Modularity Class value, and then set the node color based on the Modularity Class. We differentiated the three levels of links, genus to subgenus, subgenus to species, and species to host, by setting three different weights from high to low. We set the edge thickness according to the corresponding weight and the edge color changed with the corresponding node color.

### Recombination analysis

To investigate recombination events, we conducted recombination analysis on different subgenera of viruses, including *Decacovirus*, *Minunacovirus*, *Myotacovirus*, *Nyctacovirus*, *Rhinacovirus*, *Pedacovirus*, *Nobecvirus*, *Sarbecvirus*, *Merbecovirus*, and Unclassified Alpha-CoVs, using the RDP5 software. We employed a total of seven methods, including RDP[58], GENECONV[59], BOOTSCAN[60], MAXCHI[61], CHIMAERA[62], SISCAN[63], and 3SEQ[64], to detect potential recombination events. However, only recombination events detected by at least three of the above methods were considered true recombination events. All settings were kept to their default values, except for specifying that sequences are linear. To ensure the accuracy of the inferred recombination events, we manually checked the approximate recombination sequences and breakpoints inferred from the detected recombination events, based on phylogenetic evolutionary trees and recombination signal analysis features in RDP5, and made necessary adjustments. Due to the high number of recombination breakpoints detected, we analyzed the distribution of breakpoints to determine if there were any recombination hot- or cold-spots within the sequences analyzed. We used the Breakpoint Distribution Plot[30] for this purpose. The black line in this plot indicates the number of recombination breakpoints that fall within 200 nucleotides of the genome coordinates indicated on the *x*-axis. The light and dark gray areas represent the 95% and 99% confidence intervals of the expected degree of breakpoint clustering under random recombination. We considered genome coordinates at which the black line spikes up above the light/dark gray area to be statistically supported recombination hot-spots, and those where the black line dips below the light/dark gray area to be statistically supported recombination cold-spots.

### Prediction of host ecological distribution

We collected the corresponding bat distribution information at the IUCN (https://www.iucnredlist.org/resources/grid) (for bat species not recorded on the IUCN, we supplemented the data from GBIF (https://www.gbif.org/occurrence/search)). Meanwhile, we downloaded 19 standard bioclimatic data from Worldclim (https://worldclim.org/data). The latest available climate data layer, 19 climate variable layers (Version 2.1) related to rainfall and temperature (1970–2000) were updated in January 2020, The 19 climate variables were annual mean temperature (Bio1), monthly mean temperature variation (Bio2), isothermality (Bio3), seasonal temperature variation (Bio4), maximum temperature in the warmest month (Bio5), minimum temperature in the coldest month (Bio6), annual temperature variation (Bio7), wettest mean temperature in the wettest season (Bio8), driest mean temperature in the driest season (Bio9), mean temperature in the warmest season (Bio10), mean temperature in the coldest season (Bio11), annual rainfall (Bio12), rainfall in the wettest month (Bio13), rainfall in the driest month (Bio14), seasonal variation of rainfall (Bio15), wettest season rainfall (Bio16), driest season rainfall (Bio17), warmest season rainfall (Bio18) and coldest season rainfall (Bio19). ENMtools[65] was used to screen 19 standard bioclimatic data and mitigate the impact of

factor correlations. After removing highly correlated environmental factors, variables included several bioclimatic parameters (Bio2, Bio3, Bio4, Bio10, Bio11, Bio12, Bio14, Bio15). In addition, we incorporated other environmental factors such as elevation (https://worldclim.org), land use (http://www.gscloud.cn), and rivers (https://www.resdc.cn) to enable a more comprehensive simulation of bat distribution. The map of China, obtained from 10.5281/zenodo.4167299, and environmental factors were processed using ArcMap, including adjust coordinate system, resampling, and masking extraction, before being included in the model construction. The potential distribution of bats was predicted by maxent3.4.1 based on maximum entropy model[65,66]. 75% of the distribution points were randomly selected for model establishment, and the remaining 25% were used for model validation. Other parameters were the default parameters of the model, and the results were output in logistic format. The AUC value was used to evaluate the model prediction, all AUCs for testing and training were above 0.7. The output results of the model were imported into ArcMap 10.8 for further analysis, and the prediction results of the model were reclassified and predicted to obtain the relevant ecological adaptability evaluation map.

### Structural analysis of the RBD

Swiss-model[67] was used to model the three-dimensional structure of the receptor-binding domain (RBD) from YN2020B (OK017852), which was identified in our previous study and showed the highest sequence similarity (95.8%) to the SARS-CoV in the whole genome. The template used was PDB:2AJF.1. Subsequently, an alignment was performed between the RBD of YN2020B and the SARS-CoV-RBD-hACE2 complexes. Similarly, for the RBD of the previously identified SARS-CoV-2-related coronavirus (HN2021A: OK017803), we conducted a three-dimensional structure simulation using PDB:6M0J as the template and then aligned the simulated RBD with the SARS-CoV-2-RBD-hACE2 complexes. Finally, the amino acid residue sites where differences were observed were annotated, and images were generated using the open-source program PyMOL.

### Cross-species transmission analysis

Considering the diversity of viral sequences and the advantages of the data obtained in this study, we selected complete RDRP sequences for analysis of cross-species transmission. To ensure the reliability of our analysis, we excluded viral sequences without sampling time or unknown hosts, resulting in a total of 770 sequences used for cross-species transmission analysis, including 436 sequences belonging to *Alphacoronavirus* and 334 sequences belonging to *Betacoronavirus*. For evaluating cross-species transmission of *Alphacoronavirus* and *Betacoronavirus*, we aligned the sequences using MAFFT v7.475[52] and constructed phylogenetic trees using IQtree[53] with the most optimal models, then We used TempEst[68] to detect temporal signals and found that there was insufficient signal for both datasets, and thus we did not use Tipdates. Instead, we used a fixed substitution rate of 1.0 for all our BEAST analyses to ensure model reliability. For the subsequent BEAST analysis, we used the best-fitting model for each dataset, GTR +Empirical+Gamma for *Alphacoronavirus* and HKY+ Empirical+ Gamma+ Invariant for *Betacoronavirus*, with a strict clock model and a constant size model as a coalescent tree prior in BEAST v1.10.4[69]. The number of steps at every 1000 states was calculated by multiplying the number of sequences by 3000. We assessed the convergence of the chain using Tracer v1.7.154[70], and the effective sample sizes (ESS) values of our data were >200. Regarding the tree file generated by BEAST, we utilized TreeAnnotator v1.10.4 with a burn-in of 10% states to annotate the tree. Subsequently, we obtained an MCC (maximum clade credibility) tree, which was then annotated using FigTree for further analysis and visualization. The annotation process involved applying informative labels, colors, and other visual cues to the tree nodes and branches using FigTree[71]. This allowed us to highlight

relevant features, such as clades, branches with significant support, or other important characteristics within the tree structure. We evaluated the significance of cross-species transmission among hosts using the Bayes Factors (BF) in SpreaD3[72,73], where BF > 3 was considered a credible result, and BF > 10 was considered a highly credible result. We only present the results with BF > 3 to ensure credibility.

## Reporting summary
Further information on research design is available in the Nature Portfolio Reporting Summary linked to this article.

## Data availability
The raw data, and CoV sequence generated in this study have been deposited in the Sequence Read Archive (SRA) and Genome Sequence Archive (GSA) of the National Genomics Data Center, respectively, under accession codes PRJNA994658 and PRJCA009015. The CoV sequences generated in this study have been deposited in the NCBI GenBank with accession numbers OQ175021-OQ176213. DNA barcoding data from selected bat samples, targeting the mitochondrial cytochrome b gene, have been deposited in NCBI GenBank under accession numbers ON640659-ON640727. The protein structure data used in this study are available in the PDB database under accession code 2AJF.1 and 6M0J. The Supplementary data 1–10 are archived in Figshare, accessible via the link [https://figshare.com/s/7460ae3960c82e9f3de0] with the associated [https://doi.org/10.6084/m9.figshare.23941539].

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

## Acknowledgements

In this work, Z.-Q.W. was supported by the National Key R&D Program of China (Grant No. 2021YFC2300902), the CAMS Innovation Fund for Medical Sciences (Grant No. 2021-I2M-1-038), Beijing Natural Science Foundation (Grant No. M21002), Science & Technology Fundamental Resources Investigation Program (Grant No. 2022FY100901), the National Natural Science Foundation of China (Grant No. 32070407), the Non-profit Central Research Institute Fund of Chinese Academy of Medical Sciences (Grant No. 2019PT310029 and 2021-PT310-004), the Fundamental Research Funds for the Central Universities (Grant No. 3332021092), and the National Science and Technology Infrastructure of China (Grant No. National Pathogen Resource Center-NPRC-32). Y.-L.H. was supported by the Special Research Fund for Central Universities, Peking Union Medical College (Grant No. 3332022145).

## Author contributions

Conceived and designed the experiments: Z.W., J.W., Q.J. Performed the experiments: Y.H., P.X., Y.W. Analyzed the data: Y.H., P.X., Y.W., Z.W. Contributed reagents/materials/analysis tools: W.Z., J.Z., S.Z. Collected samples and confirmed species: W.Z., J.Z., S.Z., Z.W. Wrote the paper: Y.H., P.X., Z.W.

## Competing interests

The authors declare no competing interests.
