## [Peer Review File · Nature Communications]

REVIEWER COMMENTS

Reviewer #1 (Remarks to the Author):

This is an important work of bat sample collection, sequencing and analysis that greatly complements the available information on bat coronaviruses. Nevertheless, the manuscript is mainly descriptive. In addition, the English language is sometimes difficult to understand and should be thoroughly revised

Major comments

- Information on the precise collection sites is not available and information on the possible existence of SARS-CoV-2-like viruses near the city of Wuhan is difficult to identify
-
- No analysis of RBDs has been performed, in particular for sarbecoviruses, in order to consolidate the hypothesis of possible inter-specific transmission (cf l 595 and l 601-602 of the discussion)
- There is no information on the existence of furin sites in the discovered viruses
- l 582-285: the discussion of SARS-CoV2-like strains is not supported by any data l 608-618: there is confusion between the affinity of RBD (which is already very high in BANAL-like strains and does not need the intermediate host) and the absence of a furin site
- l 576 -579: the existence of a gradient of sequence identity of SARS CoV2 strains as one moves away from Yunnan is mentioned in the discussion but does not refer to specific results
- The methodology is not well described and need further details:
 - o How many individual swabs were pooled before sequencing? Did the authors pooled oral and rectal swabs?
 - o Did the authors obtain the whole genome of bat CoVs starting from pooled or from individual bat samples? If it was from pooled swabs, how did they verify that they were not assembling chimeric genomes? Could they please provide a detailed method for the whole genome sequencing of the different individual bat CoVs?
 - o SimPlot is not a suitable tool to predict recombination events, as it does not inform on the breakpoints coordinates. The conclusions drawn by the authors regarding recombinations are therefore not strongly supported.
 - o The analysis of cross-species transmission of CoVs between bats is only based on a small (440 bp) region of the RdRP, which limits the conclusions that can be made regarding this aspect. Why did the authors not perform the analysis on the whole genome (knowing that the RdRP gene is highly conserved and presents low specificity) or on the different recombinant fragments?
 - o Although it is mentioned several times and shown that most of the genomes result from ancient recombinations, no breakpoint research work has been done: the phylogenies are therefore based on complete genomes or whole genes, which makes their results approximate
 - o How did the authors perform the 99% identity clustering before this analysis (for example, did they keep sequences presenting a similarity >99% but belonging to different bat species)?
- The phylogenetic trees presented in the main text or in the Supplementary Data are not understandable because any sequence name is provided. We strongly encourage the authors to provide in Supplementary Data all extended phylogenies so that the reader can evaluate the robustness of the conclusions regarding the evolutionary history of bat CoVs (including recombinations).
- Please define the notion of "generally stable" regarding recombination of bat CoVs (for example in lines 286-287).
- The analysis of Sarbecovirus recombination is weak, especially the conclusions regarding the origin of SARS-CoV-2. In the SimPlot analysis, what cut-off is used when the authors say "high sequence identity"? This is a key point because the authors use these observations to speculate on the recombination events that are supposed to be at the origin of SARS-CoV-2. Can the authors clarify the assertion "abnormal sequence identity in some regions" of RaTG13 and BANAL-52 viruses?
- The spillover potential of bat CoVs is highly complex to model, and cannot be restricted to the ability of a virus to infect multiple bat species (cross-species transmission paragraph) or to recombine, as

proposed by the authors (lines 592-595). We recommend the authors to moderate their conclusions regarding the spillover potential of bat CoVs for humans based on these simple criteria (for example in lines 621-623). Overall, the authors claim several conclusions that are not supported by the data. For example, they noted that MERS-CoV spike protein is highly different from bat-related merbecoviruses but they forgot the existence of an intermediate host for MERS-CoV (lines 630-633). Similar observations can be made for Decacovirus (lines 663-664). Cross-bat transmissions do not predict the spillover potential.

- What is the frequency of co-infections at the individual bat level? Which type of co-infection did the authors identify? Can the authors provide data regarding the overlapping geographical distribution of bat species?

- How the authors selected the 371 CoVs to obtain the corresponding whole genomes? Please define the "quasi-species" term (line 126).

Minor comments:

- Please provide details of the methodology used for NGS library reconstruction and bioIT analysis (nuclease treatment, genome coverage, etc.)

- Sarbecoviruses previously published by the same team are re-used in a similar analysis (<https://academic.oup.com/nsr/advance-article/doi/10.1093/nsr/nwac213/6758518>), resulting in parts of redundancy. Of note, it has been published in National Science Review (Natl. Sci. Rev., Chinese title: 国家科学评论) which is a peer-reviewed journal published by Oxford University Press under the auspices of the Chinese Academy of Sciences.

- Several references are missing throughout the manuscript (lines 67-70, lines 546-547, etc.)

- Figure 3B: sub-lineage L11 is missing.

- Figure 6:

- o there is an incongruence between the legend of Fig. 6A written in the text (noted ORF1ab) and in the figure (noted Complete genome).

- o NC_045512 is the accession number of SARS-CoV-2 Wuhan-Hu-1 strain, and not a Bat-SARSr-CoV.

- o Problem with the positioning of several sequences: SARS-CoV-2 Wuhan does not belong to SARS-CoV-1 clade.

- o SimPlot analyses should use SARS-CoV-2 Wuhan as query.

- Can the authors hypothesize on the exchange of coronaviruses between frugivorous and insectivorous bats, as suggested in Figure 8?

- Please provide details regarding bat sampling and ethics considerations.

Reviewer #2 (Remarks to the Author):

This paper presents 339 new CoV genomes from bats in China – a substantial addition to the literature. The details are interesting, with information about recombination among SARS-CoV-2-related CoVs, and about relationships between other known pathogens (PEDV, SADS-CoV) and bat-CoVs.

With the volume of new information included, it should be worthy of publication in Nature Communications once it is adequately revised, but there is a lot more that needs to be done to improve its value, clarify the methods and analysis, and properly reflect the conclusions based on the limitations of the methods/analyses.

1) Overall the paper should try to present hypotheses that the data will test rather than just make statements about CoV diversity, abundance, etc. without support. For example, Line 66 states that CoV diversity follows bat diversity. Either there should be a citation for that, or this should be one of the hypotheses that the paper tests – does the diversity of CoVs they report correlate with the evolutionary diversity of hosts, after correcting for reporting biases. The last point is key because the sampling effort is not random or uniform, it is highly biased to where researchers assumed bats were most diverse or abundant, and where they had previously found CoVs of interest.

2) line 91 states that Yunnan is the hotspot of Beta and Alpha CoVs in bats, but again no reference, or is this a hypothesis that your new sampling can test? Why not lump all the data from your previous work, include that from WIV (Latinne et al. 2020) and other research groups and test that hypothesis?

3) Results beginning line 108 state that you merged bat samples into pools based on host species, sampling site, time of collection etc. You need to provide far more details in the methods re. pooling and identification of hosts. Please list how many samples were pooled together (min, max, mean, Standard deviation). Explain the rationale for pooling by time of sample collection (was this yearly, monthly, quarterly) and by site (area of site in Km²). Please explain in the discussion and methods the limitations of pooling for CoV detection. Surely if you pool 100 samples, you'll have less potential to pick up each CoV present than if you just pool 10 samples, given the random nature of amplification reactions. Re. Host ID, please explain the details of the DNA bar coding. You will need to explain how many samples were DNA bar-coded – was it all positive samples, and a random selection of negatives (ideal)? How were decisions made on what to barcode and what not to barcode. You also need to state how many were initially found to be inaccurately ID'd at the point of capture, so the data can be assessed for rigor of the findings

4) Paragraph beginning line 135 – please clarify – What do you mean by 'detection rate' - are you discussing the number of distinct CoVs found in different bat species here, or are you trying to infer prevalence (% of individuals infected) or abundance of CoVs (number of samples positive per species for each individual separate viral strain). Which of these is it? Additionally, Figure 2 legend needs to refer to the table where the data are (I think it's S3). Table S3 is confusing – I don't know what you calculated, and what the number of viruses/number of bats is – please clarify and explain in detail in the legend and text. At line 149 you state that certain Provinces have significantly higher detection rate, but I don't see where the statistical test is, and how this corrects for sampling biases. Full statistical analysis, details, accounting of how bias is dealt with needs to be inserted into the paper. Without more detailed statistical analysis, paragraph beginning line 152 really undermines your ability to say anything about abundance, prevalence, and which bats or provinces have higher viral diversity or infection rates, therefore where risk of spillover to livestock or people might be higher.

5) Page 9, paragraph on RdRp analysis. Please test if RdRp sequence phylogeny reflect other genes and whole genome phylogeny. This is an important question because RdRp is often used as a first screen for CoVs in wildlife sampling, and often is the only gene published, hence so many in your dataset. It is important therefore to know if RdRp reflects the way CoVs naturally diverge, or doesn't, and your dataset so far has the best opportunity to do that. It will need to be done separately for alpha and beta CoVs to account for differences in the way these two groups evolve and recombine.

6) Line 385 and on. On the recombination data for SARS-CoV-2-related CoVs, the language is very important here. The results suggest that there is a movement of recombination fragments as one virus acquires another segment from a different one, then ultimately after a number of steps, this leads to SARS-CoV-2 (e.g. "in the third stage....thus forming Banal 20-52-like strains"). I don't see how the direction of evolution/recombination can be deduced from a Simplot analysis. In any case, the way the results are written it sounds like a discussion. The results here should be revised to state that Simplot cannot explain the direction of evolution/recombination, but provides some explanation of the sharing of different genes/segments among distinct viruses. Each part of this paragraph needs to be re-written in this way.

7) Line 472 and on, Bayesian analysis to identify host switches. This section is based on RdRp sequences, so it's important to conduct the work to assess whether these reflect phylogeny of other genes or recombination segments or whole genomes (see point number 5 above).

8) Line 492 and on. The ecological analysis needs to have data on site and timing, but the samples were pooled, so it's doubtful that you will be able to distinguish samples collected at different months or sequences. In any case, details on the pooling by time of sample collection (was this yearly,

monthly, quarterly) and by site (are of site in Km²) are needed to assess whether this section is valid or not. It's also necessary to identify how sample size bias is calculated, and what the test is assessing if you don't have species prevalence data.

9) Ecological projections of CoV global distribution. This whole section (Results p. 22 to end of results section) is invalid and should be removed. Firstly, the global distribution of each bat species of each bat genus is already known. While bat species range estimates could be improved based on ecological factors (e.g. current habitat distribution based on land use and landscape mapping) it is not valid to estimate viral genus distribution based on environmental variables (i.e. climatic factors like temperature or rainfall alone that ignore host specificity and factors host distribution) alone. It seems that the current manuscript has taken the bat species known to harbor CoVs from the study, projected environmental data and assumed that bats can then exist at these sites, or that the viruses they carry in one place could exist in each place with similar environmental factors. These analytical approaches ignore biogeographic factors (oceans, mountain ranges) that might isolate bat populations and prevent viral flow. They also ignore host-specificity which we don't know much about for bat-CoVs, and are also confounded by a lack of detailed data re. sampling of sites/environment/timing because these were pooled. Unless this can all be accounted for and explained, the analysis is not valid and such approaches lack biological and mechanistic explanation that are already known for obligate CoV infections in mammalian hosts.

10) Lines 557-61 are confusing. Basically you cannot infer whether your discovery of more novel alpha-CoVs is because they are more diverse, based on the data/analysis presented. You should go back to the dataset and analyze this to test this hypothesis. Running a model that corrects for sampling bias, and assesses the number of alpha vs. beta found in your sampling, then does similar for other studies (but not combined because you used different approach, including pooling) might be able to do this. Please look into it and see if it's possible based on the pooled sampling you've done.

11) lines 561-5 – again this can't be stated unless you analyze statistically and correct for sampling sizes.

12) line 582 and on – you can't say that these viruses have limited distribution or detection rate without correcting for sample sizes. Were these bats sampled at statistically much lower numbers than the hosts of other lineages? A statistical test is necessary, and data on negative results also.

13) lines 601-2 – should cite the papers that prove these viruses can infect pigs and human cells

14) line 605 – we don't know if it only infects one species without having done a very detailed sampling and testing of all species. Statistical test that corrects for sampling bias is needed.

15) line 615 – the argument is flawed because SARS-CoV-2 did infect lots of different mammal species, but did not do this with recombination – strains from humans also infected animals. Lines 608-63 is highly speculative and should be reduced to two sentences along the following lines: "Our results suggest that SARS-CoV-2 may have undergone multiple recombination events in recent evolutionary time. This may indicate that the propensity of Sarbecoviruses to recombine is critical to their ability to emerge as human pathogens, and therefore that further surveillance of these viruses in wildlife should also consider recombination among strains as a risk factor for public health."

16) line 630 needs a citation

17) sentence beginning on line 680 overstates the ability of this analysis to predict spillover risk and should be re-written as follows: "In conclusion, this study has presented substantial new data on the occurrence of bat-CoVs in China, including a large number of novel whole genome sequences. Our results provide reference data for a deeper understanding of evolutionary relationships among a group of viruses that has caused disease outbreaks in people and livestock, including the global pandemic of

COVID-19. We have analyzed patterns of recombination, and host-virus relationships that may be able to assist in targeting surveillance and identifying key foci for spillover risk.”

Editing: The paper should be edited for English style during the revision process, including removing a number of phrases that have a subjective content, or unclear meaning:

- Line 49 ‘Sword of Damocles’)
- Stored in bats should be ‘hosted by’ bats.
- Fig 1 should list “bats known to be hosts of CoVs”
- Figure legends need much more detail. Each of the individual charts in each figure needs to be explained in full detail. If the data from a figure or chart are also given in full detail in a table, that needs to be referred to in the figure description.
- Line 277, remove ‘small-scale’ re. SADS outbreaks – they killed >25,000 pigs over multiple sites.
- Line 354, edit these two sentence to improve accuracy so that they say: “The CoVs in Sarbecovirus were mainly from the genus Rhinolophus. These include SARS-CoV and SARS-CoV-2 that can cause serious respiratory disease in people”.
- Line 370 should read “...and East Asia, but have not been reported from China (Fig. 6B)”
- line 533 should read “Expanding our understanding of the host and coronavirus diversity and distribution may yield valuable information to enhance targeted surveillance and spillover prevention programs”.
- Line 541 – remove “of 1141” – these were not all new bat-CoV strains.

Reviewer #3 (Remarks to the Author):

Han, Xu, Wang and colleagues present a large scale sequencing study of bat coronaviruses in China. Given the most recent pandemic, more coronavirus sequence data from bats are needed but as it is now the manuscript is overwhelmingly descriptive and serves more as a citation for the dataset. What cursory analyses are presented, e.g. recombination, some phylogenetics are superficial and do not reveal anything that has not been presented before for example by Latinne et al. (2020).

There's also some questionable choices made regarding the use of BEAST without tip date calibration and many instances where the authors make statements that are questionable e.g.

lines 62-63 The authors state that "The unique viral replication mechanism of bats makes them recombine frequently". There's no such thing as "a unique viral replication mechanism of bats", surely the authors mean that coronaviruses recombine frequently but that bats may provide more opportunities for them to do so.

lines 63-64 Similarly, "the genome of CoVs has strong plasticity and is prone to cross-species transmission" does not make sense. A combination of viral ecology (that supplies opportunity) and virus host/tissue tropism are what determine whether a given virus is prone to cross-species transmission, not recombination or plasticity.

lines 126-127 The sentence "we selected 371 strains as quasi-species for whole genome sequencing" makes no sense. Also, the term "quasispecies" has a specific evolutionary meaning and it is not just "viral diversity" which is what I presume the authors meant here.

In its present form the manuscript does not have much substance, is extremely monotonous, and does not present anything novel. Journals specialising in dataset publishing like Genome Announcements or the like would be a better fit for this manuscript and even then it would need significant streamlining of the text.

Response to Reviewers

Response to Reviewer #1:

This is an important work of bat sample collection, sequencing and analysis that greatly complements the available information on bat coronaviruses. Nevertheless, the manuscript is mainly descriptive. In addition, the English language is sometimes difficult to understand and should be thoroughly revised

RE: Thank you for your valuable feedback on our manuscript. We have carefully considered your comments and made the necessary revisions to address the concerns raised. we have thoroughly revised and edited the entire manuscript to improve its readability and clarity. We have made every effort to ensure that the English language is more comprehensible and to eliminate any potential ambiguities. We believe that these revisions significantly enhance the quality and accessibility of our paper.

1. -Information on the precise collection sites is not available and information on the possible existence of SARS-CoV-2-like viruses near the city of Wuhan is difficult to identify

RE: Thank you for highlighting the importance of providing more precise information on collection sites, particularly concerning the potential presence of SARS-CoV-2-like viruses near Wuhan. To address

this issue and enhance the clarity of our study, we have updated the Supplementary Table S2 to include more specific details about the sampling locations in positive samples. This addition will help clarify the geographical distribution and potential associations of the identified viruses, including those found near Wuhan.

2. -No analysis of RBDs has been performed, in particular for sarbecoviruses, in order to consolidate the hypothesis of possible inter-specific transmission (cf 1 595 and 1 601-602 of the discussion)

RE: To address your concern about the analysis of RBDs, we have not only created supplementary figures illustrating the RBDs in Sarbecovirus but also conducted additional analyses (Supplementary Fig. 6). We attempted to construct phylogenetic trees based on RBDs of 11 CoV subgenera in this study to explore the diversity of RBDs and their relationships with hosts. During the annotation of RBD regions in the 11 CoV subgenera, we found that the RBDs and their positions on the S protein were not previously characterized in the seven subgenera of Alpha-CoV. To explore the relationship between the seven subgenera of Alpha-CoV and their hosts, we adopted an alternative approach by annotating and extracting the S1 region of these subgenera to construct the evolutionary tree (Supplementary Fig. 2c). It is worth noting that the S1 region of *Rhinacovirus* has not been definitively identified and thus was not included

in the analysis. However, we successfully annotated and extracted the RBDs of the four subgenera of Beta-CoV, allowing us to construct the RBD evolutionary tree for Beta-CoV (Supplementary Fig. 2d). We believe that these additional analyses and the presented information significantly contribute to supporting our hypothesis and enhancing the understanding of potential inter-specific transmission of CoVs.

3. -There is no information on the existence of furin sites in the discovered viruses

RE: We appreciate your observation that our study lacked information on the existence of furin sites in the discovered viruses. In response to this concern, we have made significant revisions to our manuscript. Specifically, we have conducted an analysis of furin sites and incorporated the findings into the *Sarbecovirus* section of our study results. By aligning the amino acid sequences of the spike proteins from different CoVs at the subgenus level, we have examined the presence of furin sites, thereby providing insights into the origin of furin sites in SARS-CoV-2. The details of this analysis can be found in Supplementary Figure 6, located at positions lines 522-533 in the revised manuscript.

4.1 -L582-585: the discussion of SARS-CoV2-like strains is not supported by any data.line 582-585

RE: We acknowledge your concern about the need for data on the distribution and host diversity of different types of SARS-CoV-2-like strains to support our conclusions. To address this issue, we have provided the relevant data and incorporated the corresponding analyses into the *sarbecovirus* section of our study results. We have created a heatmap based on the similarity between SARS-CoV-2-like strains and SARS-CoV-2 in different genomic regions, and we have grouped strains with similar features together. Additionally, we have marked the different groups of strains on a map to visualize their host characteristics and geographical distribution patterns, lines 494-521 (Fig. 7).

4.2 - L608-618: there is confusion between the affinity of RBD (which is already very high in BANAL-like strains and does not need the intermediate host) and the absence of a furin site.

RE: Thank you for your valuable feedback, we acknowledge that the mention of BANAL-like strains may not requiring an intermediate host might weaken the persuasiveness of our argument. In light of this, we have carefully considered your suggestions and revised this section based on the additional analyses of RBDs and furin sites that we have conducted. Based on our new findings, we have emphasized the significance of the spike protein's RBD in facilitating inter-species transmission of CoVs. The revised discussion can be found at positions lines 715-725 in the revised

manuscript. We believe that these modifications can enhance the clarity and coherence of our argument.

5. -L576-579: the existence of a gradient of sequence identity of SARS-CoV2 strains as one moves away from Yunnan is mentioned in the discussion but does not refer to specific results line 576-579.

RE: We appreciate your feedback and recognize that our discussion lacked supporting data and references in this part. To address this issue, we have added information on the gradual change in sequence identity for SARS-CoV-2 strains in the map of *Sarbecovirus* section (Fig. 7). Furthermore, we have cited the article published in National Science Review (<https://doi.org/10.1093/nsr/nwac213>) as a reference to support our statement.

6-11 - The methodology is not well described and need further details:

6. -How many individual swabs were pooled before sequencing? Did the authors pool oral and rectal swabs?

RE: We apologize for not providing sufficient details on the pooling strategy. Generally, we pooled approximately 30 samples together to form a pool, with the number of samples in a pool typically not exceeding 50. We have now included a detailed description of the pooling strategy in the Methods section to address this oversight (Lines 819-832). As for your

second question, during the pooling process, we did not differentiate between oral and rectal swabs.

7. -Did the authors obtain the whole genome of bat CoVs starting from pooled or from individual bat samples? If it was from pooled swabs, how did they verify that they were not assembling chimeric genomes? Could they please provide a detailed method for the whole genome sequencing of the different individual bat CoVs?

RE: We apologize for not providing sufficient details about how we got the whole genome sequences. We have now supplemented the manuscript with a detailed explanation of the methods and strategies used to obtain the complete genomes. In summary, we first used metagenomic sequencing to obtain all CoV-related sequence information from the pooled samples, which served as a reference for further obtaining complete coronavirus genomes from individual samples. We then screened for CoVs in individual samples from the positive pools using universal primers and performed whole-genome amplification, with primers designed based on the metagenomic sequencing results.

8. -SimPlot is not a suitable tool to predict recombination events, as it does not inform on the breakpoints coordinates. The conclusions drawn by the authors regarding recombinations are therefore not strongly supported.

RE: Considering the limitations of SimPlot in analyzing recombination events, we have performed recombination analysis using 7 different methods in RDP5, as detailed in the methods section. These methods include the analysis of recombination breakpoints for 11 CoV subgenera, allowing us to obtain more precise information about the occurrence and location of recombination events. Furthermore, for the recombination analysis in the *Sarbecovirus* section, we have utilized GARD to detect recombination breakpoints. By incorporating these additional analyses, we have strengthened the reliability and robustness of our findings regarding recombination events in the revised manuscript.

9. -The analysis of cross-species transmission of CoVs between bats is only based on a small (440 bp) region of the RdRP, which limits the conclusions that can be made regarding this aspect. Why did the authors not perform the analysis on the whole genome (knowing that the RdRP gene is highly conserved and presents low specificity) or on the different recombinant fragments?

RE: We understand your concerns about using only a small fragment (440 bp) of the RdRp for the analysis of cross-species transmission. We would like to provide the following clarifications and improvements:

(1) Generally, using whole-genome data indeed provides a better representation of virus cross-species transmission. However, due to the

frequent recombination characteristics in CoVs, studying cross-species transmission based on whole-genome data is limited by the presence of recombinant sequences.

(2) Through an extensive recombination breakpoint analysis, we have found that the RdRp region, in comparison to non-structural proteins and ORFs of CoVs, exhibits a higher degree of conservation with minimal occurrence of recombination breakpoints. This observation highlights the RdRp region's relative stability and suitability for assessing host switches.

(3) While we could use recombination analysis to filter out non-recombinant sequences, we would face issues such as reduced host diversity and loss of important sequences.

(4) Given that we have identified a large number of CoV whole-genomes in our study, we can obtain more complete RdRp regions instead of being limited to the small 440 bp fragment. Therefore, considering these factors, we have chosen to use the complete RdRp region for cross-species transmission analysis to balance sequence diversity and integrity.

We hope these adjustments and explanations address your concerns and provide a more robust analysis of cross-species transmission in our study.

10. -Although it is mentioned several times and shown that most of the genomes result from ancient recombinations, no breakpoint research work

has been done: the phylogenies are therefore based on complete genomes or whole genes, which makes their results approximate

RE: Thank you for your valuable feedback. Based on the recombination breakpoint analysis mentioned earlier, we have revised the results section for *Sarbecovirus* and provided a more reliable description of the recombination events in SARS-CoV-2, in conjunction with the previously mentioned mosaic plot, rather than solely relying on phylogenetic trees. Furthermore, for the recombination analysis in the *Sarbecovirus* section, we have utilized GARD to detect recombination breakpoints (Lines 507-515). By incorporating these additional analyses, we have strengthened the reliability and robustness of our findings regarding recombination events in the revised manuscript.

11. -How did the authors perform the 99% identity clustering before this analysis (for example, did they keep sequences presenting a similarity >99% but belonging to different bat species)?

RE: As mentioned earlier, we have re-performed the cross-species transmission analysis, including sequence selection. For bat species, we retained all sequences containing complete RdRp regions from public databases and those identified in our study. For other species, particularly human and livestock sequences with close relationships, we only selected representative strains.

12. -The phylogenetic trees presented in the main text or in the Supplementary Data are not understandable because any sequence name is provided. We strongly encourage the authors to provide in Supplementary Data all extended phylogenies so that the reader can evaluate the robustness of the conclusions regarding the evolutionary history of bat CoVs (including recombinations).

RE: We understand that the phylogenetic trees presented in the main text and Supplementary Data were not easily understandable due to missing sequence labels. To rectify this issue, we have included the Accession numbers in the phylogenetic tree of main text. Then, more detailed labels were provided in the supplementary phylogenetic trees whose labels include Accession number, strain/isolate, host, and location, with appropriate abbreviations (e.g., MN996532_RaTG13_Bat/RhAf-CHN). We have also provided detailed information on the sequences in the Supplementary table S2 and S6.

13. -Please define the notion of “generally stable” regarding recombination of bat CoVs (for example in lines 286-287).lines 286-287 : Moreover, phylogenetic analysis based on the ORF1a, ORF1b, and structural proteins showed that CoVs in Rhinacovirus were generally stable, except for being jumped by WA3607r-CoVs of Decacovirus in the S protein tree.

RE: We appreciate your feedback on the notion of "generally stable" in relation to recombination of bat CoVs. We agree that it is an imprecise and vague expression. In the revised manuscript, we have rephrased this passage as follows: "Moreover, phylogenetic analysis based on the ORF1a, ORF1b, and structural proteins showed that the clustering trend of CoVs in *Rhinacovirus* did not change significantly in different evolutionary trees, except for being jumped by WA3607r-CoVs of *Decacovirus* in the S protein tree." (Lines 322-325)

14. - The analysis of *Sarbecovirus* recombination is weak, especially the conclusions regarding the origin of SARS-CoV-2. In the SimPlot analysis, what cut-off is used when the authors say "high sequence identity"? This is a key point because the authors use these observations to speculate on the recombination events that are supposed to be at the origin of SARS-CoV-2. Can the authors clarify the assertion "abnormal sequence identity in some regions" of RaTG13 and BANAL-52 viruses?

RE: Thank you for your insightful comments and questions regarding our analysis of *Sarbecovirus* recombination, particularly the conclusions related to the origin of SARS-CoV-2. We acknowledge the need for further clarification and supporting data in these aspects. Regarding the cut-off used for "high sequence identity" in the SimPlot analysis, we apologize for the confusion caused by our previous manuscript. In the revised manuscript,

we have removed the discussion on SimPlot analysis, as we recognized its limitations in accurately determining recombination breakpoints. Instead, we have incorporated more robust methods to analyze recombination events by RDP5, to obtain more precise information about the occurrence and location of recombination events. Furthermore, in response to your inquiry about the assertion of "abnormal sequence identity in some regions" of RaTG13 and BANAL-52 viruses, we have included a heatmap that illustrates the sequence identity between different regions of SCr-CoVs and SARS-CoV-2. This visualization allows for a clearer understanding of the sequence identity variations (Fig. 7).

15. - The spillover potential of bat CoVs is highly complex to model, and cannot be restricted to the ability of a virus to infect multiple bat species (cross-species transmission paragraph) or to recombine, as proposed by the authors (lines 592-595). We recommend the authors to moderate their conclusions regarding the spillover potential of bat CoVs for humans based on these simple criteria (for example in lines 621-623). Overall, the authors claim several conclusions that are not supported by the data. For example, they noted that MERS-CoV spike protein is highly different from bat-related merbecoviruses but they forgot the existence of an intermediate host for MERS-CoV (lines 630-633). Similar observations can be made for

Decacovirus (lines 663-664). Cross-bat transmissions do not predict the spillover potential.

RE: we recognize the complexity of modeling the spillover potential of bat CoVs and agree that our initial conclusions may have been overly simplistic. To address this, we have made the following revisions in our manuscript:

(1) We have carefully revised our conclusions regarding the spillover potential of bat CoVs for humans based on cross-species transmission and recombination events. We acknowledge that these factors alone cannot fully predict the spillover potential and have adjusted our conclusions accordingly.

(2) We have updated our analysis with the latest recombination and cross-species transmission results to provide more substantial evidence to support our conclusions.

(3) In the revised manuscript, we have adopted a more cautious tone when discussing our findings, emphasizing the complexity of the spillover potential and the need for further research.

(4) We have also considered the role of intermediate hosts in the transmission of viruses like MERS-CoV and have updated our discussion on lines 736-744 to reflect this information. Similarly, we have revised our observations on *Decacovirus* (Lines 765-769) to ensure that our conclusions are not overly reliant on cross-bat transmissions.

16. - What is the frequency of co-infections at the individual bat level? Which type of co-infection did the authors identify? Can the authors provide data regarding the overlapping geographical distribution of bat species?

RE: Regarding the prevalence of co-infections among individual bats, we have included additional data in our CoV screening section. Although the occurrence of co-infections in our study was relatively low, we have provided specific information on the number of co-infected bats and the corresponding types of CoVs they were found to be infected with. Comprehensive details regarding these co-infections can be found in Table S2, which presents a breakdown of the specific CoV types detected in each co-infected bat individual along with the corresponding quantitative data (Table S2). In this study, we identified the phenomenon of co-infection with coronaviruses in 63 individual samples. A total of 138 strains (identified by RdRp) were detected across these samples. The majority of co-infection type involved different subtypes of CoV within the same CoV species. Additionally, we observed a small number of co-infections involving different viral species within the same subgenus, as well as co-infections between different subgenera. As for the overlapping geographical distribution of bat species, we have previously described this information in our published article in Nature Science Review. We have

included a reference to this publication in our manuscript to provide readers with the necessary context regarding the geographical distribution of different bat species (Table S2).

17. - How the authors selected the 371 CoVs to obtain the corresponding whole genomes? Please define the “quasi-species” term (line 126).

RE: We selected the 371 CoVs for whole-genome sequencing based on the following criteria. First, we constructed a phylogenetic tree using the partial RdRp sequences obtained from our screening. We prioritized the sequences that formed distinct evolutionary branches for whole-genome amplification. Additionally, for sequences that were located on the same evolutionary branch, we selected one representative strain for whole-genome amplification. We apologize for the confusion caused by using the term "quasi-species." We will replace it with "representative strains" to better describe the diverse sequences we selected for whole-genome amplification.

Minor comments:

18. - Please provide details of the methodology used for NGS library reconstruction and bioIT analysis (nuclease treatment, genome coverage, etc.)

RE: Thank you for your suggestion. We have provided detailed descriptions of the methodology used for NGS library reconstruction and bioinformatics analysis in the Methods section of the revised manuscript.

19. - Sarbecoviruses previously published by the same team are re-used in a similar analysis (<https://academic.oup.com/nsr/advance-article/doi/10.1093/nsr/nwac213/6758518>), resulting in parts of redundancy. Of note, it has been published in National Science Review (Natl. Sci. Rev., Chinese title: 国家科学评论) which is a peer-reviewed journal published by Oxford University Press under the auspices of the Chinese Academy of Sciences.

RE: We appreciate your attention to this matter. In the revised manuscript, we have streamlined the overlapping content and focused our analysis on the recombination events in *Sarbecovirus*.

20. - Several references are missing throughout the manuscript (lines 67-70, lines 546-547, etc.)

RE: Thank you for pointing out the missing references. We have added the relevant references in the revised manuscript as suggested (Line 69 and line 649).

21. - Figure 3B: sub-lineage L11 is missing.

RE: The reason for its exclusion is that Figure 3B specifically focuses on the evolutionary analysis of the 10 bat CoV subgenera and an unclassified AlphaCoV related to China. The novel lineage L11 belongs to the BetaCoV and was discovered in Malaysia. Therefore, we did not include this sequence in the RdRp phylogenetic tree.

22-26. - Figure 6:

22. -there is an incongruence between the legend of Fig. 6A written in the text (noted ORF1ab) and in the figure (noted Complete genome).

RE: We appreciate your attention to detail. We have carefully revised the figure and its legend to ensure consistency between the text and the figure in the revised manuscript.

23. -NC_045512 is the accession number of SARS-CoV-2 Wuhan-Hu-1 strain, and not a Bat-SARSr-CoV.

RE: We apologize for the mistake in the figure. We have corrected the label for NC_045512 in the new phylogenetic tree and conducted a thorough check of all labels.

24. -Problem with the positioning of several sequences: SARS-CoV-2 Wuhan does not belong to SARS-CoV-1 clade.

RE: Thank you for pointing out the positioning error. We have carefully reviewed and corrected the labels in the revised phylogenetic tree.

25. -SimPlot analyses should use SARS-CoV-2 Wuhan as query.

RE: In response to the limitations of SimPlot analysis, we have removed the SimPlot analysis-related content from the revised manuscript.

26. -Can the authors hypothesize on the exchange of CoVs between frugivorous and insectivorous bats, as suggested in Figure 8?

RE: Regarding the hypothesis on the exchange of CoVs between frugivorous and insectivorous bats, our inferences are based on Bayesian analyses. When the Bayesian factor is greater than 3, it indicates that the host switches are reliable. We only describe the potential exchange based on the inferred results from BSSVS depended on Bayesian, which does not necessarily mean that cross-species transmission has occurred. The actual situation needs further validation. We are only providing a possibility based on model predictions.

27. -Please provide details regarding bat sampling and ethics considerations.

RE: We have included detailed information on bat sampling and ethics considerations in the revised manuscript.

We sincerely appreciate your valuable comments, which have greatly contributed to the improvement of our manuscript.

Response to Reviewer #2:

This paper presents 339 new CoV genomes from bats in China – a substantial addition to the literature. The details are interesting, with information about recombination among SARS-CoV-2-related CoVs, and about relationships between other known pathogens (PEDV, SADS-CoV) and bat-CoVs.

With the volume of new information included, it should be worthy of publication in Nature Communications once it is adequately revised, but there is a lot more that needs to be done to improve its value, clarify the methods and analysis, and properly reflect the conclusions based on the limitations of the methods/analyses.

RE : Thank you for providing valuable suggestions and feedback on our manuscript. In response to your comments, we have carefully revised and supplemented the research methods in our manuscript. Specifically, we have provided more detailed explanations of the bat species identification methods and addressed the strategies employed for sample mixing. Furthermore, we fully acknowledge the limitations present in our paper, particularly the need for better support for our conclusions. To address these concerns, we have made extensive efforts to enhance the manuscript. We have undertaken appropriate modifications and additions throughout the text to strengthen the overall reliability and validity of our findings.

Thank you again for your insightful comments, which have undoubtedly contributed to the enhancement of our work.

1. -Overall the paper should try to present hypotheses that the data will test rather than just make statements about CoV diversity, abundance, etc. without support. For example, Line 66 states that CoV diversity follows bat diversity. Either there should be a citation for that, or this should be one of the hypotheses that the paper tests – does the diversity of CoVs they report correlate with the evolutionary diversity of hosts, after correcting for reporting biases. The last point is key because the sampling effort is not random or uniform, it is highly biased to where researchers assumed bats were most diverse or abundant, and where they had previously found CoVs of interest.

RE: We understand your concern about presenting well-supported hypotheses and statements throughout the paper. Based on your feedback, we have revised our manuscript to address these concerns as follows:

(1) Regarding the statement on line 66 about CoV diversity following bat diversity, we have modified the text to reflect a more cautious approach. The revised sentence now reads: "The rich diversity of bat CoVs within China may be intricately tied to the abundant species and their widespread geographical distribution." (Lines 67-68)

(2) As for conclusions in our revised manuscript, we have made every effort to support them with data and relevant references wherever possible. We have carefully examined each conclusion and either provided supporting evidence or rephrased them as possibilities when sufficient support was lacking. This ensures that our conclusions are grounded in scientific evidence and accurately reflect the limitations of our study.

(3) In the case of potential reporting biases due to non-random or non-uniform sampling efforts, we have acknowledged this limitation in our discussion and interpretation of the results. Furthermore, we have emphasized the importance of future research efforts with more comprehensive and unbiased sampling strategies to better understand the correlation between CoV diversity and host evolutionary diversity.

2. -line 91 states that Yunnan is the hotspot of Beta and Alpha CoVs in bats, but again no reference, or is this a hypothesis that your new sampling can test? Why not lump all the data from your previous work, include that from WIV (Latinne et al. 2020) and other research groups and test that hypothesis?

RE: We appreciate your suggestion to support our statement about Yunnan being a hotspot for Beta- and Alpha- CoVs in bats. In our revised manuscript, we have made the following changes:

(1) We have incorporated data from our present and previous work, as well as data from other research groups such as WIV (Latinne et al., 2020) to provide a more comprehensive analysis of CoV diversity and distribution in different provinces of China, including Yunnan.

(2) Through our analysis of the newly identified CoV data and existing data in databases, we have observed distinct differences in the subgenus and viral species levels of CoVs among different provinces in China. Specifically, the diversity of Alpha-CoVs in Yunnan is significantly higher compared to other provinces, while the diversity of Beta-CoVs is similar to that of Guangxi and Guangdong, but noticeably higher than the remaining provinces (Fig. 5 and Fig. 6). These findings support our claim that Yunnan is a hotspot for Alpha- and Beta-CoVs in bats. To strengthen this conclusion, we have added relevant references to our revised manuscript.

3. -Results beginning line 108 state that you merged bat samples into pools based on host species, sampling site, time of collection etc. You need to provide far more details in the methods re. pooling and identification of hosts. Please list how many samples were pooled together (min, max, mean, Standard deviation). Explain the rationale for pooling by time of sample collection (was this yearly, monthly, quarterly) and by site (area of site in Km²). Please explain in the discussion and methods the limitations of

pooling for CoV detection. Surely if you pool 100 samples, you'll have less potential to pick up each CoV present than if you just pool 10 samples, given the random nature of amplification reactions. Re. Host ID, please explain the details of the DNA bar coding. You will need to explain how many samples were DNA bar-coded – was it all positive samples, and a random selection of negatives (ideal)? How were decisions made on what to barcode and what not to barcode. You also need to state how many were initially found to be inaccurately ID'd at the point of capture, so the data can be assessed for rigor of the findings.

RE: We appreciate your request for more detailed information regarding sample pooling, host identification, and the limitations of our approach. In the revised manuscript, we have provided the following clarifications:

(1) We have included detailed information on the number of samples pooled together, including the minimum, maximum, mean, and standard deviation. This information can now be found in the Methods section (lines 817-819) and supplementary Fig. 1.

(2) We have explained the rationale for pooling samples based on time of collection (e.g., yearly or monthly) and site. This information is also provided in the Methods section (Lines 819-832). In the Methods sections, we have elaborated on the limitations of pooling for CoV detection (Lines 859-862). However, we would like to emphasize that our use of

metagenomic sequencing and pooling strategies served only as an important reference for obtaining whole-genome sequence data. We further screened for the presence of CoVs in individual samples and obtained whole genome sequences of CoVs based on both single samples and metagenomic data. This approach helped mitigate the potential reduction in CoV detection when a larger number of samples are pooled together due to the random nature of amplification reactions.

(3) The identification of bat species in our study was initially based on morphology by experienced field biologists, namely Shuyi Zhang, Junpeng Zhang, and Wenliang Zhao. These biologists possess extensive expertise in bat ecology and are considered reliable in their morphological species identification. To further validate the species identification, we randomly selected a subset of bat samples for DNA barcoding. In this process, we employed the mitochondrial cytochrome b gene and utilized patagium as the tissue source. The barcoding data obtained from these samples have been submitted to GenBank (ON640659-ON640727). Importantly, the results of our DNA barcoding were found to be consistent with the morphological species identification performed by the experienced field biologists. This provides additional confidence in the accuracy of our species identification. Please note that due to the random selection of bat samples for species identification, the samples used may

not correspond directly to the distinction between positive and negative CoV samples mentioned in your query.

4. -Paragraph beginning line 135 – please clarify – What do you mean by ‘detection rate’ - are you discussing the number of distinct CoVs found in different bat species here, or are you trying to infer prevalence (% of individuals infected) or abundance of CoVs (number of samples positive per species for each individual separate viral strain). Which of these is it? Additionally, Figure 2 legend needs to refer to the table where the data are (I think it’s S3). Table S3 is confusing – I don’t know what you calculated, and what the number of viruses/number of bats is – please clarify and explain in detail in the legend and text. At line 149 you state that certain Provinces have significantly higher detection rate, but I don’t see where the statistical test is, and how this corrects for sampling biases. Full statistical analysis, details, accounting of how bias is dealt with needs to be inserted into the paper. Without more detailed statistical analysis, paragraph beginning line 152 really undermines your ability to say anything about abundance, prevalence, and which bats or provinces have higher viral diversity or infection rates, therefore where risk of spillover to livestock or people might be higher.

RE: We appreciate your request for clarification on the term "detection rate" and for more detailed statistical analysis in our study. In the revised manuscript, we have provided the following clarifications:

(1) By "detection rate," we refer to the proportion of positive CoV findings in different bat species within the same province or subgenus. For example, the detection rate of a species within a specific subgenus = the number of CoVs belonging to that subgenus identified from the species/the total number of the species. The detection rate of a species within a particular province = the number of CoVs identified from the species collected in that province/the total number of that species collected in that province (Table S3). We have added a reference to Table S3 in the legends of Figure 2.

(2) In the revised manuscript, we have included the full details of our statistical methods, including the specific statistical tests used. Regarding the specific point mentioned in line 149 about certain provinces having significantly higher detection rates, we conducted a one-way analysis of variance (ANOVA) to compare the detection rates among different provinces. However, we did not observe significant differences among the provinces, and therefore, we have removed the specific results related to higher detection rates in certain provinces as mentioned in line 149. Nevertheless, through the Nonparametric tests, we identified significant differences in the detection rates among different host families and species,

which are presented in Table S4 and Table S5. The corresponding analysis details can be found in the revised manuscript at lines 142-164.

(3) Thank you for bringing up the issue of sampling bias in our study. I fully acknowledge the presence of sampling bias and understand that it is an inherent limitation imposed by field sampling conditions. Consequently, it is inevitable that a certain degree of bias exists in our findings. In light of this, we recognize the importance of addressing and mitigating sampling bias in future research endeavors.

5. -Page 9, paragraph on RdRp analysis. Please test if RdRp sequence phylogeny reflect other genes and whole genome phylogeny. This is an important question because RdRp is often used as a first screen for CoVs in wildlife sampling, and often is the only gene published, hence so many in your dataset. It is important therefore to know if RdRp reflects the way CoVs naturally diverge, or doesn't, and your dataset so far has the best opportunity to do that. It will need to be done separately for alpha and beta CoVs to account for differences in the way these two groups evolve and recombine.

RE: We appreciate your suggestion to test if the RdRp sequence phylogeny reflects other genes and whole genome phylogeny. In response to this concern, we have conducted a comprehensive analysis in the revised manuscript to address this important question. For Dataset 1, which

consists of 475 sequences for AlphaCoVs and 408 sequences for BetaCoVs, mirroring those utilized in the 7 conserved replicase domains (CRDs) tree, we constructed phylogenetic trees using partial RdRp (~440bp) as well as other genomic regions, including ORF1ab, ORF1a, OEF1b, S, E, M, and N (Supplementary Fig.2a and 2b, Supplementary Fig. 3, and Supplementary Fig.4). By comparing the clustering patterns within these phylogenetic trees with those derived from the 7 CRDs, ORF1ab, ORF1a, and OEF1b, we observed a significant degree of congruence at the CoV Subgenus and species levels. Specifically, perfect congruence was observed at the CoV Subgenus level, while basic congruence was noted at the CoV species level. Only minor variations were detected within L8 and L9, which can potentially be attributed to the inclusion of new CoV species within these two novel lineages. However, when comparing the trees derived from the S, E, M, and N regions, we observed some divergence, suggesting potential limitations in using partial RdRp for examining the genomic structural diversity of CoVs. These additional analyses have been incorporated into the revised manuscript (Lines 218-234).

6. -Line 385 and on. On the recombination data for SARS-CoV-2-related CoVs, the language is very important here. The results suggest that there is a movement of recombination fragments as one virus acquires another segment from a different one, then ultimately after a number of steps, this

leads to SARS-CoV-2 (e.g. “in the third stage....thus forming Banal 20-52-like strains”). I don’t see how the direction of evolution/recombination can be deduced from a Simplot analysis. In any case, the way the results are written it sounds like a discussion. The results here should be revised to state that Simplot cannot explain the direction of evolution/recombination, but provides some explanation of the sharing of different genes/segments among distinct viruses. Each part of this paragraph needs to be re-written in this way.

RE: We appreciate your feedback on the recombination data for SARS-CoV-2-related CoVs and the importance of accurate language in this section. We have carefully revised the manuscript to address your concerns:

We agree that Simplot analysis alone may not provide sufficient information on the direction of evolution/recombination. Therefore, we have used RDP5 and GARD to re-analyze the recombination breakpoints and employed heatmap and maps for additional analyses. We have also revised the section in question to clarify that our analyses provide insights into the sharing of different genes/segments among distinct viruses, rather than the direction of evolution/recombination. We have re-written the parts of the paragraph that sounded like a discussion, ensuring that the revised text focuses on presenting the results of our analyses.

7. -Line 472 and on, Bayesian analysis to identify host switches. This section is based on RdRp sequences, so it's important to conduct the work to assess whether these reflect phylogeny of other genes or recombination segments or whole genomes (see point number 5 above).

RE: We understand and appreciate your concerns about the use of RdRp sequences for Bayesian analysis to identify host switches and the importance of evaluating the extent to which these sequences reflect the phylogeny of other genes, recombination segments, or whole genomes. In light of your feedback, we have further refined our analysis and approach, as described below:

(1) We acknowledge that whole-genome data generally offers a more comprehensive representation of virus cross-species transmission. However, given the frequent recombination events in CoVs, analyzing cross-species transmission based on whole-genome data is complicated by the presence of recombinant sequences.

(2) Through an extensive recombination breakpoint analysis, we have found that the RdRp region, in comparison to other non-structural proteins and ORFs of CoVs, exhibits a higher degree of conservation with minimal occurrence of recombination breakpoints. This observation highlights the RdRp region's relative stability and suitability for assessing host switches

(3) While utilizing recombination analysis to filter out non-recombinant sequences is an option, it may lead to a reduction in host diversity and the loss of crucial sequences.

(4) Taking into account the large number of CoV whole genomes identified in our study, we have chosen to employ more complete RdRp regions for our analysis, rather than limiting ourselves to the small 440 bp fragment. This approach allows us to strike a balance between sequence diversity and integrity in our cross-species transmission analysis.

8. -Line 492 and on. The ecological analysis needs to have data on site and timing, but the samples were pooled, so it's doubtful that you will be able to distinguish samples collected at different months or sequences. In any case, details on the pooling by time of sample collection (was this yearly, monthly, quarterly) and by site (are of site in Km²) are needed to assess whether this section is valid or not. It's also necessary to identify how sample size bias is calculated, and what the test is assessing if you don't have species prevalence data.

RE: We appreciate your insightful feedback regarding the ecological analysis and the potential limitations imposed by the pooling of samples. After careful consideration, we agree that our current approach to ecological analysis may not provide the most accurate representation of the

data, given the limitations associated with pooling and the lack of species prevalence data.

In light of your concerns, we have decided to remove the ecological analysis section from our study. We believe that this decision ensure that our manuscript focuses on the most robust and reliable findings, providing a solid foundation for future research in this area.

We thank you for your constructive suggestions and for helping us improve the quality of our manuscript.

9. -Ecological projections of CoV global distribution. This whole section (Results p. 22 to end of results section) is invalid and should be removed. Firstly, the global distribution of each bat species of each bat genus is already known. While bat species range estimates could be improved based on ecological factors (e.g. current habitat distribution based on land use and landscape mapping) it is not valid to estimate viral genus distribution based on environmental variables (i.e. climatic factors like temperature or rainfall alone that ignore host specificity and factors host distribution) alone. It seems that the current manuscript has taken the bat species known to harbor CoVs from the study, projected environmental data and assumed that bats can then exist at these sites, or that the viruses they carry in one place could exist in each place with similar environmental factors. These analytical approaches ignore biogeographic factors (oceans, mountain

ranges) that might isolate bat populations and prevent viral flow. They also ignore host-specificity which we don't know much about for bat-CoVs, and are also confounded by a lack of detailed data re. sampling of sites/environment/timing because these were pooled. Unless this can all be accounted for and explained, the analysis is not valid and such approaches lack biological and mechanistic explanation that are already known for obligate CoV infections in mammalian hosts.

RE: We appreciate your valuable feedback on the ecological projections section of our manuscript. We understand that our current approach to estimating viral genus distribution based solely on environmental variables may not be valid, as it does not account for host specificity, biogeographic factors, and other essential aspects that can influence viral distribution.

In response to your concerns, we have decided to remove the ecological analysis section from our study. Thank you once again for your insightful suggestions, which have greatly contributed to enhancing the quality of our manuscript.

10. -Lines 557-61 are confusing. Basically you cannot infer whether your discovery of more novel alpha-CoVs is because they are more diverse, based on the data/analysis presented. You should go back to the dataset and analyze this to test this hypothesis. Running a model that corrects for

sampling bias, and assesses the number of alpha vs. beta found in your sampling, then does similar for other studies (but not combined because you used different approach, including pooling) might be able to do this. Please look into it and see if it's possible based on the pooled sampling you've done.

Lines 557-561 On the one hand, this may be related to the greater diversity of CoVs in Alpha-CoV. On the other hand, it may be due to the diversity of Beta-CoV having been studied thoroughly. After all, the highly pathogenic SARS-CoV, MERS-CoV, and SARS-CoV-2 all belonged to Beta-CoV.

RE: We appreciate your feedback regarding lines 557-561 and the need to provide further analysis to support our hypothesis concerning the discovery of more novel alpha-CoVs. As mentioned earlier, due to the limitations of our sampling efforts, there is an inevitable sampling bias in our samples. The presence of sampling bias prevents us from drawing robust and definitive conclusions when describing the abundance of alpha- and beta-CoVs. Addressing the issue of sampling bias will be a focal point of our future research endeavors. In response to your concerns, we have revised the content in lines 557-561 to present it as a possibility rather than a definite conclusion. (Lines 656-661: Interestingly, a substantial number of novel Alpha-CoVs and Beta-CoVs were identified, with a notable predominance of the Alpha-CoV, including Decacovirus, Nyctacovirus,

and Pedacovirus. Intriguingly, no novel species of Beta-CoV were discovered in China. This observed inclination could be due to a greater diversity of CoVs within Alpha-CoV, or it may be influenced by potential sampling biases.)

11. -lines 561-5 – again this can't be stated unless you analyze statistically and correct for sampling sizes. (lines 561-565: In addition, by analyzing the host characteristics of CoVs in different subgenera, we found the natural hosts of Rhinacovirus, Myotacovirus, and Sarbecovirus mainly came from a single bat genus, Rhinolophus or Myotis, while the natural hosts of other CoV subgenera came from two or more bat genera, and formed distinct evolutionary branches based on different bat genera.)

RE: Thank you for your comment regarding lines 561-565. We would like to clarify that while we did not specifically correct for sample size in this study, the integration of viral data from publicly available databases allowed us to partially address the issue of sampling bias. We would like to emphasize that the conclusions drawn were based on a comprehensive analysis that incorporated all known and classified coronaviruses from the available databases. This approach, although not directly correcting for sample size, enabled us to consider a broader range of viral diversity and account for potential sampling biases. Furthermore, we conducted statistical analysis to support our findings and provide evidence for the

conclusions drawn. Detailed information regarding the statistical analysis (Table S9), as well as the specific results, can be found in Table S2 and S6.

12. -line 582 and on – you can't say that these viruses have limited distribution or detection rate without correcting for sample sizes. Were these bats sampled at statistically much lower numbers than the hosts of other lineages? A statistical test is necessary, and data on negative results also.

The CoVs of L-R had four characteristics: (1) it had a wide distribution in China from Southwest to Eastern areas, showing a trend of the gradual decrease in sequence identity with the SARS-CoV-2 as it moved away from the border area of Yunnan province in China; (2) it showed significant genetic diversity with different distribution regions; (3) its hosts were all *R. pusillus*; (4) its ORF1b region was more closely related to SARSr- CoVs. Therefore, The L-R CoVs must have undergone a long-term and stable spread in *R. pusillus*.

RE: We appreciate your valuable input regarding line 582 and onwards. Considering the inappropriate description of the detection rate, we have removed the related description from the revised manuscript. Instead, we have focused our study on analyzing the identified strains in comparison with the strains available in public databases. This includes analyzing their genomic structural features, recombination patterns, their

current identified locations, and their sequence similarities with SARS-CoV-2. By doing so, we aim to provide a comprehensive understanding of the identified strains and their relationship with other known strains (Lines 494-506 and lines 677-687).

13. -lines 601-2 – should cite the papers that prove these viruses can infect pigs and human cells

RE: We have now included additional references in the manuscript that demonstrate the ability of bat CoVs to infect human cells, such as WIV1 and WIV16. Specifically, we have supplemented the discussion with evidence of the potential of SADSr-CoVs, carried by bats, to directly infect pigs. This inference is primarily based on the high amino acid sequence similarity (99.22% in NTD and 98.02% in CTD) between three sequences (identified in this study) in the NTD and CTD regions of the S protein and the corresponding regions of SADS-CoV. Further details can be found in the main text, specifically in lines 708-717.

14. - line 605 – we don't know if it only infects one species without having done a very detailed sampling and testing of all species. Statistical test that corrects for sampling bias is needed.

(Line 605-607: In addition, the bat species on the evolutionary clade of the SARS-CoV and SADS-CoV are very single, and there is no case of multiple animal species like SARS-CoV-2.)

RE: In our previous manuscript, we discussed the presence of a single bat host species, *R. sinicus* and *R. affinis*, within the closest clades of SARS-CoV and SADS-CoV, respectively. However, we acknowledge that this statement may have been ambiguous. Therefore, in the revised manuscript, we have removed the related content to avoid any misunderstandings.

15. -line 615 – the argument is flawed because SARS-CoV-2 did infect lots of different mammal species, but did not do this with recombination – strains from humans also infected animals. Lines 608-63 is highly speculative and should be reduced to two sentences along the following lines: “Our results suggest that SARS-CoV-2 may have undergone multiple recombination events in recent evolutionary time. This may indicate that the propensity of Sarbecoviruses to recombine is critical to their ability to emerge as human pathogens, and therefore that further surveillance of these viruses in wildlife should also consider recombination among strains as a risk factor for public health.”

RE: We appreciate your suggestion to revise the speculative nature of our discussion on SARS-CoV-2 recombination events. We understand that

our previous statement may have been overly speculative, and we are grateful for your guidance in improving the clarity and accuracy of our argument. As suggested, in the revised manuscript, we have made significant changes to the content, reducing it to two sentences that better reflect our research findings and the associated risk factors (Lines 727-732). Considering your mention of including a reference at line L630, we understand that you may want us to retain the content L623-63. If you do recommend removing the content between lines L608 and L663, we are open to following your suggestion and make the necessary revisions accordingly. Your guidance is greatly appreciated, and we ensure that the manuscript adheres to your recommendations.

Furthermore, as previously mentioned, in the revised manuscript, we have provided more robust data support for the description of recombination events in CoVs, especially in *Sarbecovirus* through the analysis of recombination time and breakpoints using RDP5 or GARD. Additionally, we have removed the speculative discussions about the three origins of SARS-CoV-2, as we acknowledge the lack of rigor in those speculations.

16. -line 630 needs a citation

RE: Thank you for your suggestion regarding the need for a citation at line 630. In response to your recommendation, we would like to clarify

that the content primarily relies on the information derived from the data records obtained through our study and publicly available databases (Table S2, S6 and S9). We have now included a reference to the corresponding data table where the information pertaining to these sequences is documented (Line 739).

17. -sentence beginning on line 680 overstates the ability of this analysis to predict spillover risk and should be re-written as follows: “In conclusion, this study has presented substantial new data on the occurrence of bat-CoVs in China, including a large number of novel whole genome sequences. Our results provide reference data for a deeper understanding of evolutionary relationships among a group of viruses that has caused disease outbreaks in people and livestock, including the global pandemic of COVID-19. We have analyzed patterns of recombination, and host-virus relationships that may be able to assist in targeting surveillance and identifying key foci for spillover risk.”

RE: Thank you for the suggested revision to the conclusion of our manuscript. We agree that it more accurately reflects the implications of our analysis. We have revised the sentence beginning on line 886 following your suggestion: "In conclusion, this study has presented substantial new data on the occurrence of bat-CoVs in China, including a large number of novel whole genome sequences. Our results provide reference data for a

deeper understanding of evolutionary relationships among a group of viruses that has caused disease outbreaks in people and livestock, including the global pandemic of COVID-19. We have analyzed patterns of recombination, and host-virus relationships that may be able to assist in targeting surveillance and identifying key foci for spillover risk."

We appreciate your continued guidance and assistance in improving our manuscript.

18. -Editing: The paper should be edited for English style during the revision process, including removing a number of phrases that have a subjective content, or unclear meaning:

- Line 49 ‘Sword of Damocles’)
- Stored in bats should be ‘hosted by’ bats.
- Fig 1 should list “bats known to be hosts of CoVs”
- Figure legends need much more detail. Each of the individual charts in each figure needs to be explained in full detail. If the data from a figure or chart are also given in full detail in a table, that needs to be referred to in the figure description.
- Line 277, remove ‘small-scale’ re. SADS outbreaks – they killed >25,000 pigs over multiple sites.
- Line 354, edit these two sentence to improve accuracy so that they say: “The CoVs in Sarbecovirus were mainly from the genus Rhinolophus.

These include SARS-CoV and SARS-CoV-2 that can cause serious respiratory disease in people”.

- Line 370 should read “...and East Asia, but have not been reported from China (Fig. 6B)”
- line 533 should read “Expanding our understanding of the host and coronavirus diversity and distribution may yield valuable information to enhance targeted surveillance and spillover prevention programs”.
- Line 541 – remove “of 1141” – these were not all new bat-CoV strains.

RE: We are grateful for your comprehensive suggestions and guidance in improving the clarity and accuracy of our manuscript. We have carefully revised the text according to your recommendations:

(1) Removed the phrase "Sword of Damocles" from line 51.

(2) Changed "stored in bats" to "hosted by bats."(Lines 56-57)

(3) Considering the ecological relevance encompassed in the previous Fig1 and the inherent challenges in depicting coronavirus hosts, we have undertaken a revision of the figure.

(4) Added more detail to the figure legends, including explanations for each chart and relevant table references.

(5) Removed "small-scale" from line 386 regarding SADS outbreaks.

(6) Edited lines 483-485 as your suggestion. “The CoVs in *Sarbecovirus* were mainly from the genus *Rhinolophus*. These include

SARS-CoV and SARS-CoV-2 that can cause serious respiratory disease in people”

(7) lines 482-489: Due to the in-depth investigation of *Sarbecovirus* section, particularly the SARSr-CoVs, in another article in National Science Review, we have made appropriate deletions in accordance with the suggestion of another reviewer. Therefore, the specific sections you highlighted for modification have already been removed.

(8) Revised lines 634-636 as recommended: “Expanding our understanding of the host and coronavirus diversity and distribution may yield valuable information to enhance targeted surveillance and spillover prevention programs”.

(9) Removed "of 1141" from line 653.

Additionally, we have thoroughly reviewed the manuscript for English style and made necessary edits to ensure proper grammar, punctuation, and readability.

Thank you once again for your valuable feedback, which has undoubtedly enhanced the quality of our manuscript.

Response to Reviewer #3:

Han, Xu, Wang and colleagues present a large scale sequencing study of bat coronaviruses in China. Given the most recent pandemic, more coronavirus sequence data from bats are needed but as it is now the manuscript is overwhelmingly descriptive and serves more as a citation for the dataset. What cursory analyses are presented, e.g. recombination, some phylogenetics are superficial and do not reveal anything that has not been presented before for example by Latinne et al. (2020).

RE: Thank you for your valuable feedback on our manuscript. We appreciate your critical perspective and understand the concerns you have raised. We have carefully considered your comments and made revisions accordingly.

In response to your concerns regarding the descriptive nature of our manuscript, we have significantly enhanced the depth of our analyses. Specifically, we have employed additional and more reliable methods for recombination detection, breakpoint analysis, as well as included relevant analyses on the RBD (receptor-binding domain) and furin site. These additions have substantially enriched the content of our manuscript, providing a more comprehensive understanding of the bat coronavirus sequences.

We acknowledge the work by Latinne et al. (2020) and agree that their study has made important contributions to the field. In our revised

manuscript, we have taken their findings into account and have placed our study in the context of their work. Additionally, we have expanded our discussion to highlight the novel aspects and insights provided by our study, which further contribute to the existing knowledge on bat coronaviruses. We kindly invite you to review our revised manuscript and provide any further valuable suggestions. We genuinely appreciate your guidance throughout this review process. Thank you for your continued support and consideration.

1. -There's also some questionable choices made regarding the use of BEAST without tip date calibration and many instances where the authors make statements that are questionable e.g.

RE: Regarding the use of BEAST without tip date calibration: Based on our research and understanding of published studies on coronaviruses, we found that many of them did not include time signal calibration, possibly due to the relatively short sampling period of most alpha- and beta-CoVs. Although we didn't use tip date calibration in our analysis, we did use a fixed substitution rate of 1.0 for all our BEAST analyses to ensure model reliability.

2. -lines 62-63 The authors state that "The unique viral replication mechanism of bats makes them recombine frequently". There's no such

thing as "a unique viral replication mechanism of bats", surely the authors mean that coronaviruses recombine frequently but that bats may provide more opportunities for them to do so.

RE: We appreciate you for pointing out the confusion caused by our statement regarding the "unique viral replication mechanism of bats." We acknowledge that the phrase was not accurately expressed, and we apologize for the misunderstanding it may have caused.

To clarify, we intended to refer to the unique replication mechanism of coronaviruses, rather than a specific mechanism exclusive to bats. We understand the importance of precise language in scientific communication and revised the sentence to accurately convey our intended meaning. We value your feedback and improve the clarity and accuracy of our research. You can refer to lines 64-65 in the revised manuscript to find the updated statement (“The unique viral replication mechanism of CoVs makes them recombine frequently, hence the genome of CoVs demonstrates significant plasticity and may facilitate potential cross-species transmission”).

3. -lines 64-65 Similarly, "the genome of CoVs has strong plasticity and is prone to cross-species transmission" does not make sense. A combination of viral ecology (that supplies opportunity) and virus host/tissue tropism are what determine whether a given virus is prone to cross-species transmission, not recombination or plasticity.

RE: We appreciate your insightful comment regarding our statement on the plasticity of coronavirus genomes and their plasticity for cross-species transmission. We agree that factors such as viral ecology and virus-host/tissue tropism play crucial roles in determining the likelihood of cross-species transmission. We acknowledge the need for clarification in our original statement. We would like to highlight that our analysis of different subgenera of coronaviruses, which have previously experienced cross-species transmission, revealed a higher frequency of recombination events. While we understand that cross-species transmission is a complex process influenced by various factors just like what you said, our study simply observes this correlation and suggests the existence of a potential unknown association between recombination events and cross-species transmission, which merits further investigation. We have made the necessary revisions to ensure greater accuracy in our expression. Please refer to line 64-65 in the revised manuscript to find the updated statement.

4. -lines 126-127 The sentence "we selected 371 strains as quasi-species for whole genome sequencing" makes no sense. Also, the term "quasispecies" has a specific evolutionary meaning and it is not just "viral diversity" which is what I presume the authors meant here.

RE: Thank you for bringing up the issue regarding our statement about selecting 371 strains as "quasi-species" for whole-genome

sequencing. We apologize for the confusion caused by using the term "quasi-species." We replaced it with "representative strains" to better describe the diverse sequences we selected for whole-genome amplification. Please refer to line 133 in the updated version for the precise modification.

5. -In its present form the manuscript does not have much substance, is extremely monotonous, and does not present anything novel. Journals specialising in dataset publishing like Genome Announcements or the like would be a better fit for this manuscript and even then it would need significant streamlining of the text.

RE: We appreciate your concern about the novelty of our research. In this study, we identified 863 strains of CoVs and obtained 330 whole-genome sequences. Among the newly identified strains, at least one new virus species is unique to them, involving multiple unclassified virus species. We acknowledge the value of our genomic data but recognize the need for more in-depth analysis to obtain novel results. Based on your feedback and the suggestions from other reviewers, we have adopted new analytical methods and strategies to re-analyze our genomic data, including sample bias correction, evolutionary analysis, recombination breakpoint analysis, cross-species transmission analysis, and sequence structural

features (e.g., RBDs, furin site). We have also improved the language and presentation of our manuscript.

We hope that our responses and revisions address your concerns and demonstrate the significance and novelty of our research. Thank you for your valuable feedback.

REVIEWER COMMENTS

Reviewer #1 (Remarks to the Author):

The authors have improved the manuscript, taking into account most of my concerns. The revised manuscript is clearer and improved, especially regarding the following points:

- The methods are now well described, especially regarding the pooling and whole genome sequencing strategies.
- The authors addressed one of our main concerns regarding the recombination analysis, using state-of-the-art tools (RDP5) instead of Simplot. The additional figure 7 is very important and informative, and adds significant value to the manuscript.
- The conclusions of the article are now in adequation with the results, and are not over-interpreted as it was for example for the spillover potential of these new viruses.

We need still few information:

- Genomic structure of the new viruses: how was made the functional annotation of the genomes (lines 237-258)?
- Similarly, did the authors performed functional analyses (= cleavage) to determine the biological role of the furin motif identified in some of their Embecovirus, Merbecovirus, and Hibecovirus genomes?

To improve the new version of the manuscript, we suggest the authors the following modifications:

- Can the authors prepare a map showing the geographical distribution of bat species in China, in light of the abundance of coronaviruses they found? Is there any overlap between bat richness in a specific area and high CoV abundance in that specific area?
- The designation of new lineages as L1, L2... is hard to follow: based on ICTV demarcation criteria, can the authors propose a new taxonomic classification of these lineages (new sub-genus, new species...) and provide a tentative name for these new lineages?
- Figure 7 is very important. But why the authors analyzed recombinations compared to SARS-CoV-2 only and not to both SARS-CoV-2 and SARS-CoV-1? The phylogenetic positioning of their sarbecoviruses with regards to SARS-CoV-1 and SARS-CoV-2 is not well defined.
- We think that the information of the possible co-evolution of their viruses with specific bat species or genera (lines 662-664), and the date of divergence between viruses shared by different species would be an added value of the article, as there is few information regarding these aspects in the literature. However, it is not a critical point.
- Can the authors speculate on the ability of their sarbecoviruses to use ACE2 to enter human (or mammalian) cells according to the analysis of contact residues (from line 702)? Again, this is not a critical point, but it would enrich the article.

Reviewer #2 (Remarks to the Author):

The authors have addressed all of my previous comments adequately, in particular by removing sections of the discussion that are not supported by the data or analysis, and tightening up the methods so that the work can be further analyzed in the future. While the paper is by nature descriptive, I believe that the data are very important and will be a very widely read and cited paper. I recommend accept at this point, but with the proviso that the English grammar and the style of the text is improved by Nat. Comm. editors.

Reviewer #3 (Remarks to the Author):

Despite some streamlining of the manuscript in response to my previous comments, Han, Xu, and Wang added more analyses that then necessitate more descriptive language to explain. I remain

unconvinced that Nature Communications is the correct journal for what is essentially a manuscript describing a dataset with exploratory analyses attached and which will be primarily cited for the sequence data rather than any findings arising out of them.

The analyses as they are now, including the additional ones, are somewhat standard so there's not much to comment about their validity. The issue I raised about BEAST use was more to do with applying a molecular clock to a dataset that is not only recombinant but also collected over time, potentially resulting in subtrees that are incorrectly rooted because of the latter (not too much of an issue) and meaningless because of the former.

An additional issue I failed to notice before is that though the assembled contigs have been uploaded to GenBank the underlying raw reads were placed on a separate platform. I believe such practices are detrimental to science - they split scientific databases and can eventually allow anyone in control of such non-standard databases to withhold raw data whenever it suits them.

Response to Reviewers

Response to Reviewer #1:

The authors have improved the manuscript, taking into account most of my concerns. The revised manuscript is clearer and improved, especially regarding the following points:

- The methods are now well described, especially regarding the pooling and whole genome sequencing strategies.
- The authors addressed one of our main concerns regarding the recombination analysis, using state-of-the-art tools (RDP5) instead of Simplot. The additional figure 7 is very important and informative, and adds significant value to the manuscript.
- The conclusions of the article are now in adequation with the results, and are not over-interpreted as it was for example for the spillover potential of these new viruses.

RE: We sincerely appreciate your recognition of the improvements that we have incorporated into our manuscript following your invaluable feedback. Your insightful comments were crucial in guiding these improvements. In response to your latest comments, we have carefully considered and made additional revisions accordingly.

1. We need still few information:

1.1 - Genomic structure of the new viruses: how was made the functional annotation of the genomes (lines 237-258)?

RE: We appreciate your inquiry into the functional annotation of the new viruses identified in our study. As described in the "Genome assembly and annotation" section of our Methods (lines 927-938), we primarily utilized the 'Annotate & Predict' tool of the Geneious Prime software for this task. In instances where annotations using NCBI's RefSeqs were found to be inadequate, we pinpointed sequences from public databases that demonstrated significant alignment with our viral sequences and subsequently used these for re-annotation.

Post initial annotation, we implemented a rigorous cross-checking process with well-annotated sequences from the same CoV subgenus available in public databases, utilizing the MAFFT plugin in Geneious Prime for sequence reordering and alignment. Special attention was given to the accuracy of the start and end points of each gene, with a specific focus on the proteins encoded by these genes.

Significantly, we noticed anomalies in some sequences, primarily involving abnormal start and end points of genes or irregular elongation or premature termination of the proteins they encode. These anomalies were suspected to be caused by sequencing errors. Consequently, primers were specially designed for PCR amplification targeting these anomalous regions. The subsequent sequencing data derived from these amplified products were then utilised to rectify these identified anomalies.

It is important to clarify that our functional annotation approach primarily depends on bioinformatics analysis. In situations where anomalies were observed, like abnormal gene start and end points or irregular protein termination/elongation, we did utilize PCR-based experimental validation for corrections. However, we acknowledge that beyond these targeted confirmations, we haven't conducted further functional experiments to validate our annotations, and we recognize this as an area for potential improvement.

1.2 - Similarly, did the authors performed functional analyses (= cleavage) to determine the biological role of the furin motif identified in some of their Embecovirus, Merbecovirus, and Hibecovirus genomes?

RE: We appreciate your interest in the functional analysis of the furin motifs identified in the *Embecovirus*, *Merbecovirus*, and *Hibecovirus* genomes. In our study, we utilized the ProP method for the computational prediction of furin cleavage sites in coronaviruses. This method, available through the ProP server, employs a neural network-based approach trained on a dataset of known cleavage sites. The neural network is designed to recognize the specific patterns in amino acid sequences that indicate the presence of a cleavage site. This method has been validated and demonstrated high accuracy in predicting proprotein convertase cleavage sites, making it a reliable tool for our computational study. For a more

detailed understanding of the ProP method, we refer to the original publication by Duckert, Brunak, and Blom ^[1].

We acknowledge that our study does not include experimental validation of the biological role of the predicted furin motifs. We agree that functional analyses, such as cleavage assays, would provide a more comprehensive understanding of the biological significance of these motifs. However, we would like to clarify that the scope of the current study is confined to computational predictions. While we recognize the importance of experimental validation, it is beyond the scope of our current work. We appreciate your suggestion and will consider incorporating functional analyses in our future research to complement our computational predictions.

2. To improve the new version of the manuscript, we suggest the authors the following modifications:

2.1 - Can the authors prepare a map showing the geographical distribution of bat species in China, in light of the abundance of coronaviruses they found? Is there any overlap between bat richness in a specific area and high CoV abundance in that specific area?

RE: We appreciate your insightful suggestion. Accordingly, we have updated our study and now provide a visual representation of bat species distribution and coronavirus abundance in Supplementary Fig 2. For

modelling the geographical distribution of coronavirus-associated bat species in China, we amalgamated bat distribution data from the IUCN and GBIF databases. This data was further nuanced with the integration of various environmental influencers, including climate parameters, altitude, land use, and river systems. To enhance the accuracy of our model, we excluded highly correlated environmental variables as determined by ENMtools. We employed the Maximum Entropy Model-based software, MaxEnt3.4.1, to carry out the simulation predictions for bat distribution. The model exhibited robust performance, with AUC values for both the training and test sets surpassing 0.7. The stratification of bat distribution was achieved with the aid of ArcMap10.8. We overlaid the geographical coordinates of coronaviruses identified in this study on the Chinese map. Interestingly, our findings suggest that regions with a higher concentration of bats present an elevated likelihood of detecting coronaviruses. Readers can find further details on these methodologies in the specified section, commencing from line 1036-1068.

2.2 - The designation of new lineages as L1, L2... is hard to follow: based on ICTV demarcation criteria, can the authors propose a new taxonomic classification of these lineages (new sub-genus, new species...) and provide a tentative name for these new lineages?

RE: We deeply appreciate your insightful comments regarding the nomenclature of our newly discovered lineages. In response to your suggestion and after a comprehensive reevaluation of the taxonomic characteristics inherent to the new lineages of coronaviruses, we have reformulated their nomenclature. We adopted "Host+Coronavirus Taxonomic Category+Lineage Identifier" as our new naming standard for these lineages. Nevertheless, given the significant coronavirus species level diversity among bat hosts within these lineages, we opted for a format that embraces this diversity while retaining consistency. Thus, our revised naming convention is "Bat+Genus/Subgenus+Lineage Identifier". An example of this would be Bat Alphacoronavirus New Lineage 1 (BatAlpha_NL1).

2.3 - Figure 7 is very important. But why the authors analyzed recombinations compared to SARS-CoV-2 only and not to both SARS-CoV-2 and SARS-CoV-1? The phylogenetic positioning of their sarbecoviruses with regards to SARS-CoV-1 and SARS-CoV-2 is not well defined.

RE: We greatly appreciate your recognition of the importance of Figure 7. Regarding your inquiry as to why our recombination analysis was restricted to SARS-CoV-2 and did not include SARS-CoV-1, it is indeed a complex issue. In our understanding, though SARS-CoV-1 and SARS-

CoV-2 exhibit considerable genomic divergence, they are still categorized into the same coronavirus species based on the ICTV classification criteria. Given the existence of potential recombinant strains, the accurate distinction and definition between SARS-CoV-1 and SARS-CoV-2 can be challenging. In our study, we conducted a recombination analysis of the whole Sarbecovirus subgenus, but a detailed recombination analysis was only performed on SC2r-CoVs, which have been described in past research as having a close phylogenetic relationship with SARS-CoV-2 in part or the entire genome structure, and they form a separate cluster from SC1r-CoVs on the ORF1ab phylogenetic tree.

Three reasons underpin our decision not to extend the recombination analysis to SARS-CoV-1:

(1) As you noted in your previous review, our team has already conducted a deep exploration into Sarbecoviruses. Therefore, in this revised manuscript, we significantly reduced parts overlapping with our previous work and focused on presenting the newest and most significant findings of this study, namely, the speculation of the SARS-CoV-2 production process.

(2) Despite identifying numerous recombination events in SC1r-CoVs, we did not observe a process resembling our proposed SARS-CoV-2 production. In our previous work, we performed an analysis of sequence consistency on various gene fragments of Sarbecoviruses with SARS-CoV-

1 and SARS-CoV-2 and represented this through heatmaps. This, in part, resonates with your suggestion regarding the recombination analysis of SARS-CoV-1.

(3) Moreover, the recombinant origin of SARS-CoV-1 has been thoroughly analyzed in a paper published by Hu. B et al. in 2017 ^[2], where they identified all gene segments constituting SARS-CoV-1 in Bat SARSr-CoVs and proposed a hypothesis for the recombinant origin of SARS-CoV-1. This fact was another major factor behind our decision not to delve further into the recombination analysis of SARS-CoV-1.

We hope the explanation above adequately addresses your concerns.

2.4 - We think that the information of the possible co-evolution of their viruses with specific bat species or genera (lines 662-664), and the date of divergence between viruses shared by different species would be an added value of the article, as there is few information regarding these aspects in the literature. However, it is not a critical point.

RE: We acknowledge your suggestion for further exploration of the co-evolution of bats and their corresponding viruses. We attempted to use eMPRes software to perform such an analysis, but encountered significant difficulties given the large number of animal species (over 100) and the approximately 800 coronaviruses involved. Even after reducing redundancy by selecting only those coronaviruses with at least 90%

consistency in RdRp sequences, around 200 coronaviruses remained. This extensive diversity led to confusing results from the eMPRESS software that were difficult to interpret. Consequently, we chose to forego this approach and instead utilized Maximum Clade Credibility (MCC) tree analysis at the subgenus level, which allowed us to track host shifts throughout the coronavirus diversification process more effectively. This approach yielded results that were clearer and more aligned with your expectations. We have included these findings in our revised manuscript and believe they add substantial value to our work (line 651-662).

2.5 - Can the authors speculate on the ability of their sarbecoviruses to use ACE2 to enter human (or mammalian) cells according to the analysis of contact residues (from line 702)? Again, this is not a critical point, but it would enrich the article.

RE: Following your advice, we selected two virus strains from the sarbecoviruses identified in our research for further analysis (line 556-573). Specifically, we focused on YN2020B (OK017852), which exhibited the highest sequence similarity to SARS-CoV across the entire genome (nt, 95.8%). We used Swiss-model to model the three-dimensional structure of the RBD from YN2020B with PDB:2AJF.1 serving as the template. Upon successful modeling, we aligned the simulated RBD of YN2020B with the SARS-CoV-RBD-hACE2 complex. In parallel, we conducted a similar

process for another identified SARS-CoV-2-related coronavirus strain HN2021A (OK017803) identified in our study. We simulated its RBD structure using PDB:6M0J as the template and aligned the modeled RBD with the SARS-CoV-2-RBD-hACE2 complex.

Through these alignments, we identified divergent amino acid residues at the contact interface with hACE2, in comparison to SARS-CoV and SARS-CoV-2 RBDs. We annotated these divergent residues and visualized the changes using the open-source PyMOL program. We believe that this additional analysis, highlighting the specific amino acid residue differences at the hACE2 contact interface, not only addresses your query but also enriches our manuscript.

References

1. Chica, R.A., Protein engineering, design and selection. 2020, Oxford University Press. p. gzaa024.
2. Hu, B., et al., Discovery of a rich gene pool of bat SARS-related coronaviruses provides new insights into the origin of SARS coronavirus. PLoS pathogens, 2017. **13**(11): p. e1006698.

Response to Reviewer #2:

The authors have addressed all of my previous comments adequately, in particular by removing sections of the discussion that are not supported by the data or analysis, and tightening up the methods so that the work can be further analyzed in the future. While the paper is by nature descriptive, I believe that the data are very important and will be a very widely read and cited paper. I recommend accept at this point, but with the proviso that the English grammar and the style of the text is improved by Nat. Comm. editors.

RE: We would like to express our heartfelt gratitude for acknowledging our efforts in addressing your valuable comments and for recognising the potential impact of our study. Your suggestion of refining the language and style of the manuscript is duly noted. In response to this, we have conducted further revisions to enhance the clarity and precision of the text. As suggested, we are looking forward to the editors of Nature Communications bringing their expertise into the refinement of grammar and style, ensuring the work is presented in its best form.

Thank you once again for your invaluable input.

Response to Reviewer #3:

Despite some streamlining of the manuscript in response to my previous comments, Han, Xu, and Wang added more analyses that then necessitate more descriptive language to explain. I remain unconvinced that Nature Communications is the correct journal for what is essentially a manuscript describing a dataset with exploratory analyses attached and which will be primarily cited for the sequence data rather than any findings arising out of them.

RE: Thank you for your insights and the time you have dedicated to reviewing our manuscript. You are correct in pointing out that our work, at the initial stage, was more descriptive. However, it is our belief that this level of description was crucial to lay down a foundation for the readers and to set the context for our novel findings. In response to your comments and those of other reviewers, we have extensively revised our manuscript to better emphasize the analytical and novel aspects of our work. The result is a stronger paper with a clearer focus on our unique contribution to the field, underpinned by the inclusion of a comprehensive and well-structured methodology, the elaboration of our significant results, and the expansion of our conclusions. We believe that our study, with its current improvements, offers significant insights into the understanding of bat coronaviruses.

The analyses as they are now, including the additional ones, are somewhat standard so there's not much to comment about their validity. The issue I raised about BEAST use was more to do with applying a molecular clock to a dataset that is not only recombinant but also collected over time, potentially resulting in subtrees that are incorrectly rooted because of the latter (not too much of an issue) and meaningless because of the former.

RE: Thank you for your insightful comments regarding the use of BEAST in our analysis. We concur with your observation concerning the limitations of applying a molecular clock to a dataset that is recombinant and spans multiple collection times. Despite our comprehensive coronavirus sampling and our efforts to collect sequences with temporal data, we encountered the same challenge as other teams performing coronavirus analyses using BEAST: Datasets lacked sufficient temporal information to accurately estimate either substitution rates or the time to the most recent common ancestor. Consequently, we used a fixed substitution rate of 1.0 for all our BEAST analyses and scaled branch lengths according to relative time units (clock rate = 1.0) in MCC tree. We acknowledge this as a methodological limitation; however, it still provides a degree of insight into cross-species transmission events in bats. We appreciate your feedback, which will undoubtedly inform our future research endeavors.

An additional issue I failed to notice before is that though the assembled contigs have been uploaded to GenBank the underlying raw reads were placed on a separate platform. I believe such practices are detrimental to science - they split scientific databases and can eventually allow anyone in control of such non-standard databases to withhold raw data whenever it suits them.

RE: Thank you for expressing your concerns about the storage of CoV sequences and Raw data on different platforms. We understand your point about potential fragmentation of scientific databases and the hypothetical risk of data withholding.

We would like to clarify that the placement of our data in GenBank and GSA adheres to the recommended public data repositories listed by the Nature journal series (<https://www.nature.com/sdata/policies/repositories>). Although GenBank and GSA are different platforms, each adheres to the journal's policy for data storage, ensuring the scientific rigor and accessibility of our work.

The rationale behind our choice of GSA for raw data storage over GenBank was primarily due to the substantial volume of our raw data, which approximates to 764GB. GSA allowed us to upload our data more quickly and still guaranteed unrestricted public access and free downloading.

In response to your concern, we have now also uploaded the raw data to GenBank's Sequence Read Archive (SRA), under the BioProject accession **PRJNA994658** at: <https://dataview.ncbi.nlm.nih.gov/object/PRJNA994658?reviewer=1bpsk5cvf7shfefsh1v23tfdpq>. This move is aimed to foster a more integrated data access experience, ensure transparency, and maximize the potential of our findings for global scientific exploration.

We appreciate your valuable feedback and look forward to any additional comments you may have.

REVIEWERS' COMMENTS

Reviewer #1 (Remarks to the Author):

The authors have taken my comments into account and have completed their manuscript accordingly. I now consider it acceptable for publication.

Response to Reviewers

Response to Reviewer #1:

The authors have taken my comments into account and have completed their manuscript accordingly. I now consider it acceptable for publication.

RE: We sincerely appreciate your diligent review of our manuscript and the time you devoted to providing valuable feedback. We are pleased to learn that you find our revisions satisfactory and consider the manuscript suitable for publication. Your insightful comments have greatly contributed to the enhancement of our work. We are grateful for your guidance and expertise throughout this process.